



# On the retrieval of snow grain morphology, the accuracy of simulated reflectance over snow using airborne measurements in the Arctic

Soheila Jafariserajehlou[1], Vladimir V. Rozanov[1], Marco Vountas[1], Charles K. Gatebe[2,3] and John P. Burrows[1]

[1]Institute of Environmental Physics, University of Bremen, Bremen, Germany
[2]Universities Space Research Association (USRA), Columbia, MD, USA
[3]NASA Goddard Space Flight Center, Greenbelt, MD, USA

Correspondence to: Soheila Jafariserajehlou (jafari@iup.physik.uni-bremen.de)

**Abstract.** Accurate knowledge of the reflectance from snow/ice covered surface is of fundamental importance for the investigation and retrieval of snow parameters and atmospheric constituents. This is a prerequisite for identifying and quantifying changes in the environment and climate in Polar Regions. However, the current differences between simulated and measured reflectance in a coupled snow-atmosphere system, leads to systematic errors in the determination of the amount of trace gases, aerosol and cloud parameters from space based and airborne passive remote sensing observations.

In this paper, we describe studies of the retrieval of snow grain morphologies, also called habits, and their use to determine reflectance and test the accuracy of our radiative transfer model simulations of reflectance by comparison with measurements. Firstly we report on a sensitivity study. This addresses the requirement for adequate a priori knowledge about snow models and ancillary information about the atmosphere; the objective being to minimize differences between measurements and simulation. For this aim we use the well-validated phenomenological radiative transfer model SCIATRAN. Secondly and more importantly, we present a novel two-stage snow grain morphology (i.e. size and shape of ice crystals in the snow) retrieval algorithm. We then describe the use of this new retrieval to estimate the most representative snow model, using different types of snow morphologies, for the airborne observation conditions, performed by NASA's Cloud Absorption Radiometer (CAR). The results show that the retrieved ice crystal shapes are consistent with the expected snow morphology (estimated from temperature information) in the measurement area over Barrow/Utqiaġvik, Alaska in 2008.

Thirdly, we present a comprehensive comparison of the simulated reflectance (using retrieved snow grain size and shape as well as independent atmospheric data) with that from airborne CAR measurements in the visible and NIR wavelength range. The results of this comparison are used to assess the quality and accuracy of the radiative transfer model in the simulation of the reflectance in a coupled snow-atmosphere system.

Assuming that that snow layer consists of ice crystals with aggregates of 8 column ice habit having an effective radius ~ 98.83 μm, we find that for a surface covered by old snow, the Pearson correlation coefficient, R, between measurements and



simulations to be 0.98 ($R^2$ ~ 0.96). For freshly fallen snow on areas having surface inhomogenity, the correlation is ~ 0.97 ($R^2$ ~ 0.94) in the infrared and 0.88 ($R^2$ ~ 0.77) in the visible wavelengths assuming that snow layer consists of aggregate of 5 plate ice habit with effective radius ~ 83.41 µm. Largest differences between simulated and measured values are observed

in glint, i.e. in the angular regions of specular and near-specular reflection, with relative azimuth angles <± 40° in forward scattering direction. The absolute difference between the modeled results and measurements in off-glint regions with viewing zenith angle less than 50° is generally small ~ ±0.025 and does not exceed ±0.05. These results will help to improve snow surface reflectance in the snow-atmosphere system algorithms designed to retrieve atmospheric parameters such as aerosol optical thickness in the Polar Regions.

**1 Introduction**

The extent and type of snow and ice cover have a significant impact on climate, as noted by Arrhenius over 100 years ago. There is a positive feedback between decreasing surface temperature, an increase of snow and ice cover and an associated increase in planetary albedo, which then further decreases surface temperature and vice versa. Consequently, changes in snow and ice extent and morphology play a role in climate change (Schneider and Dickinson, 1974; Curry et al., 1995;

Cohen et al., 2014; Kim et al., 2017; Wendisch et al., 2017; 2019). During the past recent decades the Arctic region has warmed more rapidly than other regions. This phenomenon is known as the Arctic Amplification AA (Serreze and Barry, 2011). The analysis of the growing number of long-term records of the data products, retrieved from passive and active satellite observations, provides potentially invaluable information to identify and quantify the evolution and consequences of AA (Wendisch et al., 2017). Because of the magnitude of the scattering from snow, the use of remote sensing measurements

above snow covered surfaces in the cryosphere, requires accurate models of the scattering and reflectance from snow surfaces to retrieve information about atmospheric constituents in particular clouds and aerosol parameters but also trace gases and avoid systematic errors (e.g. Istomina et al., 2010; 2012; Jafariserajehlou et al., 2019).

A large number of experimental and theoretical studies have been conducted measuring and modeling snow optical properties such as angular distribution of reflected light over and within the snow surface and the subsequent derivation of

snow albedo. The early measurements by Middleton and Mungal (1952) and the model of Dunkle and Bevans (1956) used to analyze the transmittance and reflectance of snow cover were the beginning of considerable efforts on this topic. Barkstrom (1972) formulated and solved the scattering problem for snow surfaces in terms of radiative transfer theory. Later, substantial progress in our understanding of the angular distribution of snow reflectance has been made by comparing the simulated reflectance from snow covered surfaces calculated by Radiative Transfer Models (RTM) with observations (e.g.

Wiscombe and Warren 1980; Warren et al., 1998; Arnold et al., 2002; Painter and Dozier, 2003; Kokhanovsky and Zege, 2004; Li and Zhou, 2004; Hudson et al., 2006; Hudson and Warren, 2007; Lyapustin et al., 2010; Kokhanovsky and Breon, 2012).



The reflection/scattering patterns of snow surface can be summarized as follows: i) snow is not a Lambertian reflector in the visible and near infrared spectral region; its reflectance has anisotropic nature and the anisotropy increases with wavelength; ii) unlike other surface types e.g. vegetation or soil with a strong peak in backscattering direction (the hot spot effect), snow has a strong forward peak for large viewing zenith angles (e.g. Gatebe and King, 2016); iii) The snow reflectance variation is larger in the principal plane, i.e. the plane containing the Sun, surface normal and observation direction, than in the cross plane, i.e. the one perpendicular to principal plane (Warren, 1982; Lyapustin et al., 2010; Kokhanovsky and Breon, 2012). However, the remaining discrepancies between simulated results and field measurements led to further investigations in the field of single scattering properties of snow grains (Mishchenko et al., 1999; Jin et al., 2008, Yang et al., 1998; 2003; 2013), surface roughness (Warren et al., 1998; Hudson et al., 2006; Hudson and Warren, 2007; Lyapustin et al., 2010; Zhuravleva and Kokhanovsky, 2011) and atmospheric correction methods (Lyapustin et al., 2010). Despite substantial improvements, the uncertainties in our understanding of the microphysical and macroscopic properties of snow are an unresolved issue for RTMs, ray-tracing and climate models. For example, the current state of the art RTMs yield much more anisotropic reflectance behavior for snow in the glint region than observed in reality (Zhuravleva and Kokhanovsky, 2011; Lyapustin et al., 2010; Hudson and Warren, 2007; Warren et al., 1998). These studies either focus on the snow reflectance at the surface employing an atmospheric correction method (Leroux et al., 1998; Kokhanovsky and Zege, 2004; Kokhanovsky et al., 2005; Lyapustin et al., 2010; Negi and Kokhanovsky, 2011;) or consider the atmospheric effects without in-depth investigations of the surface parameters (Aoki et al., 1999; Hudson et al., 2006; Kokhanovsky and Breon, 2012). A comprehensive study and investigation of both snow layer and atmosphere parameters in a coupled snow-atmosphere system has not yet been undertaken but is required to improve the accuracy of remote sensing retrieval algorithms for aerosol and cloud in the Arctic region (Istomina et al., 2010; 2012; Jafariserajehlou et al., 2019).

Consequently, the goal of this study is to i) study the sensitivity of scattering and reflectance in the coupled snow-atmosphere system taking into account both surface and atmospheric parameters; ii) retrieve the most representative ice crystal morphology by applying a snow grain size and shape retrieval algorithm to measured reflectance at the wavelength of 1.6 µm; iii) evaluate the ability of a phenomenological RTM, to reproduce the measured reflectance over the spectral range 0.34 µm - 1.649 µm at all available observation directions using the retrieved atmospheric and snow parameters.

The RTM SCIATRAN (Rozanov et al., 2014) which is a well validated phenomnologicl RTM, and the Airborne observations of the scattered and reflected solar radiation, acquired by Cloud Absorption Radiometer (CAR) were used in this study. The CAR measurements were made during the Arctic Research of the Composition of the Troposphere from Aircraft and Satellite (ARCTAS) campaign over Barrow/Utqiaġvik, Alaska, in 2008. The information about the atmospheric parameters during measurement campaign of the CAR instrument was taken from available AERONET and satellite data.

The rest of this paper comprises the following. In the next section we present the theoretical background and terminology used to calculate the angular distribution of reflectance in a snow-atmosphere system. In sect. 3 and 4, the measurements and the simulation methods are introduced and explained. In sect. 5 the sensitivity of reflectance to the underlying snow layer and atmospheric parameters are investigated. In sect. 6, the results of applying the two-stage snow grain size and shape





retrieval algorithm are presented. In sect. 7, the results of the reflectance simulations are compared to CAR measurements. Finally, conclusions are drawn in sect. 8. Appendix contains detailed description of the snow grain size and shape retrieval algorithm used in the study.

## 100   2 Theoretical background

To describe the directional signature of reflectance over different surface types, the Bidirectional Reflectance Distribution Function (BRDF) as defined by Nicodemus (1965), is the commonly used reflectance quantity. The term BRDF describes the reflection of incident solar radiation from one direction to another direction (Nicodemus 1965). The mathematical form of BRDF is expressed as (Nicodemus et al., 1977; Schaepman-Strub et al., 2006):

$$BRDF_\lambda = \frac{d\,L_r(\theta_i, \varphi_i, \theta_r, \varphi_r; \lambda)}{d\,E_i(\theta_i, \varphi_i; \lambda)} \left[ sr^{-1} \right], \tag{1}$$

where $L_r$ is the reflected radiance, $\theta$ and $\varphi$ are the zenith and azimuth angles, respectively. The subscript $i$ corresponds to the incident and r to the reflected beams. $E$ is the incident surface flux (irradiance) and $\lambda$ is the wavelength. However, the BRDF is not a directly measurable quantity because of its being formulated as a ratio of infinitesimal quantities (Nicodemus et al., 1977; Schaepman-Strub et al., 2006). Nicodemus et al. (1977) provided an extensive description of reflectance terminologies and measurable quantities e.g. the Bidirectional Reflectance Factor (BRF), the Hemispherical Directional Reflectance Factor (HDRF), the Directional Hemispherical reflectance (DHR), etc. According to Nicodemus et al. (1977) and Schaepman-Strub et al. (2006) each of the terms is defined for the specific illumination and reflectance geometries for which, the reflectance properties are measured e.g. satellite, airborne or laboratory measurement conditions. Following the method of Gatebe and King (2016), the effective BRDF at a horizontal (flat) reference plane is defined as:

$$BRDF_\lambda^e = \frac{\Delta L_r(\theta_i, \varphi_i, \theta_r, \varphi_r; \lambda)}{\Delta E_i(\theta_i, \varphi_i; \lambda)} = \frac{\Delta L_r(\theta_i, \varphi_i, \theta_r, \varphi_r; \lambda)}{\Delta L_i(\theta_i, \varphi_i; \lambda)\cos\theta_i \Delta\omega_i} \left[ sr^{-1} \right], \tag{2}$$

where $BRDF_\lambda^e$ is as an average of the BRDF over an appropriate area, angle and solid angle for specific observation geometry; $\Delta\omega_i$ is a finite solid angle element. The validity of this approximation relies on the experimental evidence that the BRDF is not significantly influenced by the following effects: the finite intervals of area, angle, solid angle and the distribution function; sub-surface scattering; radiation parameters such as wavelength and polarization, fluorescence etc. (i.e. significant variations do not occur within small intervals, see Nicodemus et al., 1977; Gatebe and King, 2016). As a result, the $BRDF_\lambda^e$ is determined by:

$$BRDF_\lambda^e = \frac{L_r^e(\theta_i, \theta_r, \Delta\varphi)}{F_{0,\lambda}\cos\theta_i}, \tag{3}$$

where $L_r^e$ is the measured radiance, $F_{0,\lambda}$ is the solar irradiance incident at the top of atmosphere (TOA). Often, it is helpful to have a description of the difference between the measured surface reflectance and a Lambertian reflector; in such a case the

equivalent Bidirectional Reflectance Factor $BRF_\lambda^e$, which is $BRDF_\lambda^e$ multiplied by $\pi$ is more representative.



To isolate the reflectance properties of surface and derive $BRF_\lambda^e$ or $BRDF_\lambda^e$ just above the surface, we need to apply atmospheric correction methods on measured radiance at TOA or flight altitude e.g. by using knowledge of the atmospheric scattering or absorption using RTMs. This removes from the measured radiance, the four atmospheric contributions from the atmosphere at TOA or flight altitude (Schaepman-Strub et al., 2006): i) the atmospheric path radiance, ii) the scattering by

the atmosphere before the solar radiation has reached the surface, iii) the scattering by the atmosphere after being reflected by the surface, iv) the scattering by the atmosphere before and after reaching the surface. However, most of the atmospheric contributions in measurements close to the surface are negligible (except diffuse component number ii) and measured quantities represent the "at surface" radiance (Schaepman-Strub et al., 2006). Sensitivity studies have demonstrated that atmospheric contributions to the CAR channel observations range from 3 to 12% depending on wavelength (Soulen et al.,

2000). Consequently, previous studies presented either the BRFs in a surface-atmosphere system at flight altitude without atmospheric correction (Soulen et al., 2000) or the BRFs right above the surface after atmospheric correction (Gatebe et al., 2005; Gatebe and King 2016).

In this study, to avoid uncertainties arising from different assumptions being part of the atmospheric correction methods, no such correction is applied to measured radiances $L_{r,h}$ at flight altitude h. Instead, the reflectance at flight altitude in the

snow-atmosphere system is calculated by the equation:

$$R = \frac{\pi\, L_{r,h}(\theta_i, \theta_r, \Delta\varphi)}{F_{0,\lambda} \cos\theta_i} \tag{4}$$

where $L_{r,h}$ is the measured radiance at flight altitude. All reflectance/$BRF_\lambda^e$ values at flight altitude, presented in this study are calculated using Eq. 4 and referred to as "reflectance". In the simulation of the reflectance in a coupled snow-atmosphere system, to account for atmospheric effects contribution properly, independent data about atmospheric parameters (Aerosol

Optical Thickness (AOT) and gases absorption) at the time and close to the location of measurements are needed and taken from ground-based and space-borne measurements and applied to the simulation. More details of the atmospheric data are discussed in sect. 3 and 4. To estimate $BRF_\lambda^e$ just above the surface, further atmospheric correction is needed. We assume at infrared wavelengths where atmospheric scattering is negligible, the reflectance at flight altitude is a good estimation of $BRF_\lambda^e$ just above the surface.

## 3 Measurements


CAR is an airborne instrument, developed at NASA's Goddard Space Flight Center. It has been used during several field campaigns around the world since 1984 up to present to measure the single scattering albedo of clouds and the bidirectional reflectance of various surface types etc. For this study, we used CAR data from the ARCTAS campaign conducted at Elson Lagoon, near Barrow/Utqiaġvik, Alaska, in April 2008 (Lyapustin et al., 2010; Gatebe and King, 2016). The goal of

ARCTAS was to study physical and chemical processes in the Arctic atmosphere and surface. Date, location, measurement geometry and available atmospheric parameters during the measurements used in this study are presented in Table 1.





The unique design of CAR provides simultaneously both up-welling and down-welling radiances at 14 spectral bands located in the atmospheric window regions of UV, visible and near-infrared from 0.34 µm to 2.3 µm comprising important wavelengths relevant for remote sensing applications such as aerosol retrievals. Through a rotating scan mirror, the

instrument provides viewing geometries suitable for measurements needed for BRF calculation. CAR collects data bya mirror rotating 360° in a plane perpendicular to the direction of flight through a 190° aperture that allows acquiring data from local zenith to nadir or horizon to horizon with an angular resolution of 1°. The high angular/spatial resolution of 1° in both viewing and azimuth angles allowed the anisotropy of the reflectance in the snow-atmosphere system to be estimated with high accuracy. The spatial resolution of CAR depends on the flight altitude e.g. 10 m² and 18 m² at nadir for 600 m and

1000 m flight altitude, respectively, which increases with the viewing zenith angle (VZA) e.g. 580 m² at 80° VZA for 1000 m flight altitude. The capability of acquiring data at different altitudes (~ 200, 600 and 1700 m) enables us to evaluate the sensitivity of reflectance with respect to atmospheric effects in RTM simulation.

Examples of calculated reflectance values using Eq. (4) from CAR measurements on 7[th] of April, 2008 at Elson Lagoon (71.3° N, 156.4° W) are shown in Fig. 1 and Fig. 2. In spite of the influence of the atmospheric scattering and absorption, the

general features of the snow BRF are clearly observable. The latter comprise: i) the decrease of snow reflectance with increasing wavelength due to the increasing absorption by snow at longer wavelengths; ii) the increase of the snow BRF as a function of VZA and the strong forward scattering peak in the principal plane at large VZA; iii) the smaller angular variation of the BRF at cross plane compared to the principal plane. The reflectance values increase with altitude. The snow surface inhomogeneity decreases with increasing altitude due to the change of spatial resolution with altitude (Gatebe and King,

2016; Lyapustin et al., 2010). Accordingly, at poorer spatial resolution, spatial homogeneity are more efficiently averaged.

To account for aerosols, we use the Aerosol optical thickness, AOT data acquired by the nearby Aerosol Robotic Network (AERONET) sun-photometer at Barrow/Utqiaġvik during the CAR measurement time. AERONET is a globally distributed network and provides long-term and continuous ground-based measurements of the total column aerosol optical thickness derived from the attenuation of sun light and provided often at high temporal resolution of 15 minutes. AERONET AOT data

are provided at 0.5 µm and 0.6 µm wavelengths. We use the Ångström exponent to calculate AOT values at the reference wavelength (0.55 µm) required for the SCIATRAN simulation. Table 1 shows the calculated AOT at 0.55 µm based on AERONET data for Barrow/Utqiaġvik at the closest time to the CAR airborne measurements.

To account for ozone absorption, we use knowledge of the ozone total column amount retrieved from the space borne measurements by using the University of Bremen weighting function DOAS (WFDOAS) algorithm version 4 (Weber et al.,

2018). The WFDOAS data are selected using the criteria of having smallest temporal and spatial differences with CAR data. For nitrogen dioxide, we use vertical column information from the SCIATRAN database obtained from a 2D chemical transport model developed at University of Bremen (Sinnhuber et al., 2009).

The derived AOT and trace vertical column have been used in the simulation of radiative transfer processes in the snow-atmosphere system.



## 4 Simulations

SCIATRAN is a software package for radiative transfer modeling, developed at the Institute of Environmental Physics, University of Bremen (Rozanov et al., 2002; 2014) and freely available at http://www.iup.uni-bremen.de/sciatran/. The SCIATRAN package has been used in a variety of remote sensing studies to simulate radiative transfer processes in the spectral range from the ultraviolet to the thermal infrared (0.18 µm - 40 µm), assuming either a plane parallel or a spherical atmosphere (Rozanov et al., 2014).

To calculate reflectance values, SCIATRAN assumes that the snow is a layer with an optical thickness of 1000 and a geometrical thickness of 1 m composed of ice crystals of different morphologies and placed above a black surface. This assumption was successfully validated by Rozanov et al. (2014). The snow layer is assumed to be vertically and horizontally homogeneous and composed of a monodisperse population of ice crystals. The impact of impurities in the snow e.g. dust, black carbon etc. is neglected in this study. To simulate the radiative transfer through a snow layer, the single scattering properties of ice crystals including extinction and scattering efficiencies, single scattering albedo and phase functions need to be defined in SCIATRAN. All these parameters are dependent on the wavelength, size and shape of the particle (Leroux et al., 1999). Recently, a new data library of basic single scattering properties of ice crystal habits developed by Yang et al. (2013) has been incorporated in the SCIATRAN model (Pohl et al., Personal communication). This database comprises a full set of single scattering properties at wavelengths from the UV to the far IR for the following eight ice crystal morphologies: droxtal, column and hollow column, aggregate of eight columns, plates, small aggregate of five plates, large aggregate of ten plates, and hollow bullet rosettes. More detailed information about the ice crystal shapes and sizes can be found in Yang et al. (2013). In addition to the above-mentioned eight ice crystals, optical parameters for triadic Koch fractal (referred as fractal in this paper) particles are used as well (Macke et al., 1996; Rozanov et al., 2014). The fractal particle model uses regular tetrahedrons as its basic elements. In this study, the second generation fractals as described in Macke et al. (1996) and Rozanov et al. (2014) are utilized.

In SCIATRAN, the snow grains are specified by their single-scattering properties of sparsely distributed particles. Namely, the snow grains are assumed to be in the far field zones of each other and will thus scatter the light independently. For a snow layer, the snow grains can be located in each other's near-field, resulting in interactions between the scattered electromagnetic fields of neighboring particles which leads to modification of single-scattering properties (Mishchenko, 2014; Mishchenko, 1994). The impact of near-field effect was investigated in Pohl et al. (2019) using the modified single scattering properties of sparsely distributed particles as suggested in Mishchenko (1994). The comparison of snow BRFs calculated assuming sparsely or densely packed snow layers shows that the maximum difference does not exceed 0.015% (Pohl et al., 2019). Therefore, this effect was ignored in radiative transfer calculations through the snow layer.

To account for atmospheric effects, SCIATRAN incorporates a comprehensive database containing monthly and zonal vertical distribution of trace gases e.g. $O_3$, $NO_2$, $SO_2$, $H_2O$, etc., spectral characteristics of gaseous absorbers, vertical profiles of pressure and temperature and molecular scattering characteristics (see Rozanov et al., (2014) for details). To account for



scattering and absorption by aerosols over snow in SCIATRAN, the optical characteristics of aerosol particles and vertical distribution of aerosol number density are required. In this study we use Moderate Resolution Imaging Spectrometer

(MODIS) collection 5 aerosol parameterization (Levy et al., 2007) as an internal database in SCIATRAN. Levy et al. (2007) developed a framework for connecting the aerosol micro-physical properties such as the refractive index and size distribution to the AOT at 0.55µm. Using AOT from ground based measurements of AERONET at Barrow/Utqiaġvik as mentioned and selecting one of aerosol types, the Mie code incorporated into SCIATRAN is employed to calculate aerosol extinction and scattering coefficients. In this study, the vertical profile of aerosol number density as an "exponential vertical distribution"

for a height of 3.0 km is used.

For the conditions described above, the radiative transfer calculations are performed at a source-target-sensor geometry extracted from the airborne measurements at solar zenith angle of 70.23°, 69.11°, 67.68° and 62.11°; viewing zenith angle 0° - 70° and relative azimuth angle 0° - 360° with an angular resolution of 5° and at four different altitudes of 181 m, 206 m, 647 m and 1700 m. More detailed information about atmospheric and snow layer parameters are given and discussed

separately in the following section.

## 5 Sensitivity of reflectance to the snow morphology and atmospheric parameters

The measured reflectance in the visible and NIR spectral range over a snow field depends on the relative importance of the absorption and scattering radiative transfer processes in the atmosphere and snow layer. In this section, we investigate the sensitivity of the reflectance on the radiative transfer through the atmosphere and the snow at the selected wavelength bands:

i) 1.649 µm because of the high sensitivity of this wavelength to snow grain properties; and ii) 0.677 and 0.873 µm wavelengths because of the relatively high and differing sensitivities at these wavelengths to the atmospheric conditions and being used for aerosol optical thickness retrievals.

### 5.1 Impact of snow: size and shape of ice crystals

To study the influence of ice crystal morphology on the radiation field above snow covered surfaces, we perform the

simulation at 1.649 µm for three important reasons (Leroux et al., 1998):

i) the absorption of ice crystals is small or negligible at the selected wavelengths in the visible domain of the spectrum. In contrast, in the near-infrared range, due to the large absorption of ice crystals at these wavelengths, the snow reflectance is significantly affected by the snow grain size; the larger the particle, the smaller the reflectance because of larger absorption and stronger forward scattering;

ii) the BRF properties of snow at 1.649 µm are closer to that for single scattering behavior and it is linked to the phase matrix which strongly depends on the shape of ice crystals;

iii) the impact of the atmosphere (absorption by $CO_2$ and $H_2O$ and diffuse incident irradiance) at 1.649 µm is negligible.

To illustrate the high sensitivity of radiation field to the varying size of ice crystals at 1.649 µm, we simulated snow reflectance at principal and cross planes assuming nine ice crystal morphologies with varying sizes (here size refers to





maximum dimension/edge length) 60 ~ 10000 μm and three different roughness (smooth surface: 0, moderate surface roughness: 0.03, severe surface roughness 0.5), for further information see Yang et al. (2013). Fig. 3 shows the simulated reflectance versus the VZA in the principal plane (as the most sensitive and representative direction for the largest changes of reflectance) using severely roughened morphology. As can be seen in Fig. 3, the reflectance strongly changes with the size of ice crystals from 60 to 10000 μm. Differentiating between various shapes has the largest effect in forward scattering

(φ=0°) and lesser effect in backward scattering direction (φ=180°). The results indicate that the effect of changing size is larger than the impact of differentiating between various shapes of ice crystals at this wavelength.

Using the "aggregate of 8 columns" shape and changing maximum dimension from 60 μm to 10000 μm result in reflectance decrease of ~ 40 % at nadir (VZA ~ 0°) and more than 80 % in forward scattering direction (at VZA of 60°) which is considerably large. Changing the shape to the droxtal at the same size, provides a noticeable decrease of ~ 30% in

reflectance at forward scattering direction for a viewing zenith angle of 60° and leads to a much weaker forward peak. Noteworthy is, that the plate shape cannot reproduce the enhancement in backward direction (typical for a BRF over snow). Using aggregate of 5 and 10 plates leads to larger reflectance in all directions. However, the analysis of simulation results at cross plane (not shown here) indicates that, the impact on the reflectance pattern, originating from the specific shapes of the ice crystals is relatively small compared to the impacts at principal plane.

The large range of changes of the reflectance when using different ice crystal sizes in both the principal and cross planes highlights the importance of having accurate priori knowledge or estimation of size of the ice crystals and their shapes to simulate accurately measurements. In our study, due to the lack of such information from in situ measurements, we estimate the size of ice crystals for each selected crystal shape separately to have a priori knowledge of ice crystal properties and limit the differences between the simulated and measured reflectance. The detailed explanation and results are given in sect. 6.

**5.2 Impact of atmosphere: scattering and absorption by aerosol and gases**

The incident radiation on the snow layer is composed of direct sunlight and the diffuse radiation from the sky (Aoki et al., 1999). To take the atmospheric absorption and scattering into account, we assume an atmosphere over the snow layer, which contains: i) Rayleigh scattering (scattering by air molecules), ii) gaseous absorption and iii) absorption and scattering by aerosols. Therefore, in this section, absorption bands e.g. 0.677 μm are selected to evaluate the impact of atmosphere. We

calculate the reflectance at 0.677 μm under three different conditions, assuming a model atmosphere governed: i) by Rayleigh scattering; ii) identical to i) but with absorption by ozone ($O_3$) and nitrogen dioxide ($NO_2$); iii) identical to ii) but including aerosol. The calculations are performed assuming following properties of the atmosphere and snow layer:

i) Vertical profile of nitrogen dioxide, pressure and temperature are selected according to a 2D chemical transport model (Sinnhuber et al., 2009) incorporated in SCIATRAN;

ii) Total vertical column of ozone as well as AOT are set according to Table 1;

iii) Snow layer is composed of ice crystals having the shape "aggregate of 8 column", maximum dimension of 650 μm and severely roughened crystal surface.





Fig. 4 shows the impact of the atmosphere and the difference between measured and simulated reflectance values at three different altitudes: 206, 647 and 1700 m; for the 3 scenarios. The reflectance reduction at 647 m flight altitude due to

gaseous absorption is the smallest ~ 5% close to the nadir region and becomes larger ~ 10% in forward scattering direction which decreases to ~ 8% at 1700 m altitude. At this wavelength, ozone with vertical optical depth (VOD) of $1.6\times10^{-2}$ has a much larger contribution to gaseous absorption as compared to that of $NO_2$ with VOD of $3.95\times10^{-5}$.

The reflectance for an atmosphere containing three types of aerosol (weakly/moderately/strongly absorbing aerosol) and without aerosol (containing only Rayleigh and gaseous absorption) are presented in Fig. 4. For more information on aerosol

typing used in this study please see Levy et al. (2007). The changes in reflectance due to weakly absorbing aerosol with an AOT of 0.11 (measured by AERONET) at 206 m flight altitude are ~ 5% at nadir and increase in forward scattering direction to ~ 13%. The strongly absorbing aerosol (at the same AOT of 0.11) reduces the reflectance by ~7% at nadir and ~ 20 % in forward scattering direction. At 1700 m the reflectance decreases by 6% at nadir and 7% in forward scattering direction. The differences between the three aerosol types does not lead to changes in reflectance, which are larger than 5%

in or close to nadir areas. In summary, an atmosphere containing Rayleigh scattering, absorption by ozone ($O_3$) and nitrogen dioxide ($NO_2$) and weakly absorbing aerosol is the best representation of the atmospheric conditions for our case study.

## 6 Retrieval of snow grain size and shape

In the previous section, we showed that having adequate a priori information about snow surface and atmosphere is necessary to calculate reflectance of sufficient accuracy. In contrast to the atmospheric parameters available from

independent sensors and models, a priori knowledge about ice crystal size and shape for the underlying snow layer is not typically available. To estimate the optimal ice crystal morphology we used a snow grain size and shape retrieval algorithm, by minimizing the difference between the measured and simulated reflectance (See appendix A for details). Here size refers to effective radius[1] of the ice crystal. The retrieval algorithm is applied to measurements at principal and cross planes at 1.649 µm assuming different shape and crystal surface roughnesses. To find the best representative shape and size, the bias

and Root Mean Square Error (RMSE) between the measured and simulated reflectance were determined for each case study.

Fig. 5 shows one example of the comparison between measured and simulated reflectance at principal and cross planes. Based on comparison, one can state that the angular reflectance pattern of the CAR measurement on the 7[th] of April 2008 at Elson Lagoon is reproduced by SCIATRAN successfully. The highest accuracy is obtained by assuming ice crystals as "an aggregate of 8 columns" with severely roughened crystal surface at an effective radius of 98.8 µm (corresponding to

maximum dimension of 650 µm). In this case, the largest and smallest discrepancies appear in the forward scattering direction and close to nadir (VZA< ± 25°), respectively. The overall RMSE and bias between measurements and simulation at principal plane is 6.9 % and 2.7 % respectively. A lesser degree of agreement between simulated results and measurements are provided by using "column" and fractal shapes with an RMSE of 7.3% and 9.75%, respectively. The

---

1    effective radius= $3/4 \times(V_{tot}/A_{tot})$; $V_{tot}$: total volume and $A_{tot}$: the total projected area of ice per unit volume of air (Baum et al., 2014).





largest difference between measurements and simulations is observed for the case using "droxtal" shape with an RMSE ~

320    25.54%.

We also retrieved effective radius of ice crystal using CAR data for fresh fallen snow on the 15[th] of April 2008. Due to the existing surface horizontal inhomogeneity for the case of fresh snow acquired at lower flight altitude ~ 181 m, larger differences between simulated and measured reflectance are expected, as compared to the old snow case on the 7[th] of April 2008. The results are shown in Fig. 6. Unlike the old snow case presented in Fig. 5, the "aggregate of 8 columns" shape does

not optimally represent the ice crystals of this particular day. Rather, a reflectance simulated by using an "aggregate of 5 plates" as the ice crystal shape provides the minimum RMSE ~ 12.85% between measurement and simulation results. "Aggregate of 10 plates" and fractal provide the second and third most representative shapes with an RMSE of ~ 13.16 % and 14.69 %, respectively. The results obtained by using the "droxtal" ice crystal shape exhibit large differences in both of forward and backward scattering directions with RMSE of 34.1 %.

Though the real nature of ice crystal shape at the time of measurement is not known to us, the impact of temperature on morphology of snow grain particles has been debated in previous studies. The results of such studies are now compared with our findings (Slater and Michaelides, 2019; Shultz, 2018; Libbrecht, 2007; Bailey and Hallett, 2004; Yang et al., 2003). Based on the relationship between temperature and snow grain morphology, the column-based shapes are the dominant ice crystal morphology in environments with temperatures higher than -10°C whereas plates are dominant if the temperature is

less than -10°C. Our findings with respect to the most representative shape for each case study agree with this argument. The temperature range during CAR measurements at 6-7[th] of April 2008 is from -20 to -5°C. Based on our results the "aggregate of 8 columns" is the most representative shape for measurements conducted on this day. On 15[th] of April 2008 when the temperature range changes to -23 to -17°C, mainly plate-based ice crystal shapes are expected for such low temperatures and our results confirm this argument. In addition, the existence of droxtal ice crystals during the measurement is less probable

because very low temperatures (~ -50°C) are needed to form droxtal or quasi-spherical ice crystals (Yang et al., 2003). The temperature dependence of the ice crystal morphologies explains in part why droxtal shaped ice crystals do not capture the derived snow reflectance values from CAR measurements in any of our scenarios. With respect to size of ice crystals, we do not compare fresh and old snow cases because it is important to note that the date of old snow case is before fresh snow. This means the studied old snow case is not the aged fresh snow case. Therefore, the change of ice crystal size with its age is not

studied in the scope of this paper.

A summary of retrieved effective radii using different ice crystal shapes and corresponding bias and RMSE values is presented in Table 2. The ice crystals with minimum RMSE value at 1.649 μm are underlined and selected to be used for subsequent calculations of reflectance at 0.677, 0.873 μm. In Fig. 7, the importance of ice crystal shape selection for the snow grain size retrieval and the snow reflectance calculation is highlighted. The measurements were selected from the old

snow and fresh snow cases. The effective radius is retrieved only at 1.649 μm and then has been used to calculate the reflectance at 0.677 μm and 0.873 μm. The results are presented in Fig. 7 with corresponding RMSE and bias values in the principal plane. It can be seen that the retrieved effective radius value changes from shape to shape. The difference in



retrieved effective radius generally does not exceeds 40 % but in the case of the plate ice crystals the retrieved effective radius is ~70 % smaller than the other shapes e.g. aggregate of 8 columns. This is a significantly large difference. However,

these results are presented for the principal plane where the maximum differences between simulation and measurement is expected. Therefore, the overall bias and RMSE value on all azimuth direction is smaller than presented here. It can be seen that the RMSE values at 0.677 µm and 0.873 µm are significantly smaller than that at 1.649 µm. This is explained by the high reflectance values at these wavelengths and therefore the larger denominator in RMSE formula, in which the difference of measured and simulated reflectance is divided by measured reflectance.

## 7 Comparison of measured and simulated reflectance

In this section we present results of the comparison of measured and simulated reflectance in the snow-atmosphere system. The simulations, which used the results and findings described in the based on the previous section were performed: assuming an atmosphere containing $O_3$, $NO_2$, weakly absorbing aerosol as described in Table 1. The snow layer is assumed to be comprised of "aggregate of 8 column" ice crystals with a maximum dimension of 650 µm (effective radius 98.83 µm)

for the case of old snow, and "aggregate of 5 plates" ice crystals with a maximum dimension of 725 µm (effective radius 83.41 µm) for the case of fresh snow. To assess the accuracy of simulations over all azimuth angles, the correlation plot and the Pearson correlation coefficient between measured and modeled reflectance are shown in Fig. 10.

In Fig. 8, the difference between the simulated and measured reflectance at 0.677 and 1.649 µm is small on average, being less than 0.025 in regions of small VZA and not exceeding ±0.05 for larger VZA < 50°. These values are larger at 0.873 µm;

the maximum difference reaches ~ ±0.05 for small VZA. The difference between SCIATRAN simulation values and those of the measurements is pronounced in the forward scattering region where |Δφ| < 40°. As it is shown in Fig. 10, the correlation coefficient between reflectance measurements over old snow and simulation is high, ~ 0.98. Fig. 9 is the same plot as Fig. 8 but for fresh snow. We consider that surface inhomogentities and related larger shadowing effects at measurement altitude of 181 m, explain why the correlation decreased to 0.88 at 0.677 µm (for the case of old snow, acquired at 647 m flight altitude,

surface inhomogenities are smoothed and therefore the old snow case is less affected by surface inhomogenities). However, in Fig. 10, at 1.649 µm correlation coefficient is high ~ 0.97 for the case of fresh snow, possibly because of their being less sensitivity of this channel to shadowing and atmospheric scattering effects. In addition, the high correlation coefficient at 1.649 µm and small discrepancies < ±0.025 in off-glint region confirms the suitability of the selection of the best representation for ice crystal shape in previous step. The differences between SCIATRAN simulations and CAR

measurements of reflectance are less pronounced in the glint region, as compared to those for the old snow.

## 8 Conclusion

In this study, our objective was to assess the accuracy of the simulation of the reflectance in a snow-atmosphere system taking different snow morphology and atmospheric absorption and scattering into account. For this we used a state of the art RTM, SCIATRAN, which used explicit models of the snow layer and the airborne CAR measurements.





The airborne CAR data were acquired by NASA over Elson Lagoon at Barrow/Utqiaġvik, Alaska, during the ARCTAS campaign in spring 2008. The spectral coverage of the airborne measurements is wide (0.3-2.30 μm) comprising important wavelengths relevant for remote sensing applications such as aerosol retrievals which could benefit from the results of this study. Measurements obtained at different flight altitudes (~ 200, 600 and 1700 m) provide an opportunity to investigate the sensitivity of simulated reflectance to atmospheric parameters.

The SCIATRAN RTM (a phenomenological RTM) was used to simulate the reflectance in the snow-atmosphere system and its changes for different snow morphologies (i.e. snow grain size and shape). These simulations take atmospheric scattering and absorption explicitly into account. We investigated the sensitivity of reflectance in the snow-atmosphere system to snow grain size and shape. We have shown that the selection of the most representative shape and size of the nine ice crystals used in SCIATRAN to describe the snow surface is essential to minimize the difference between simulations and

measurements.

        To obtain a priori knowledge of snow morphology, we use the snow grain size and shape retrieval algorithm and apply it to CAR data. In our case study at Barrow/Utqiaġvik, the simulated reflectance assuming ice crystals with aggregate composed of 8 columns shape agreed well with measurements for the old snow case, having RMSE of 6.9 % and average bias of 2.7 % with respect to the measured CAR reflectance in the principal plane where the largest discrepancies are

expected. For the case of freshly fallen snow, an aggregate of 5 plates shape was the most representative ice crystal having RMSE values of 12.8 % and a bias of 11.23 % with respect to the measured CAR reflectance. The data for the freshly fallen snow case were acquired at 181 m. Larger differences as compared to the older snow case at 647 m are attributed to surface inhomogenity. The surface inhomogenity most likely originate from sastrugi. Simulation, in which the snow layer was comprised of ice crystals with a droxtal shape (being semi-spherical particles) did not yield accurate reflectance for the

snow-atmosphere system in any of our case studies. We showed that using the knowledge from studies of the temperature dependence of ice crystal morphologies agrees with our findings with respect to the most representative ice crystal size and shape for our case studies.

        In our study, the simulated patterns of the reflectance with respect to spectral and directional signatures produce well the measurements, as evidenced by the high correlation coefficients in the range of 0.88 ~ 0.98 between measurements (old and

fresh snow) and simulation at the selected wavelengths of 0.677, 0.873 and 1.649 μm. In the off-glint regions $|\Delta\varphi| > 40°$ and VZA < 50°, the overall absolute difference between the modeled reflectance from SCIATRAN and CAR measurements is below 0.05. This absolute difference in off-glint area is smaller in the short wave infrared as compared to visible. It should be noted here that the reflectance of the snow is lower in the short wave infrared compared to the visible.

        In summary, the approach shows the high accuracy of the phenomenological SCIATRAN RTM in simulating the radiation

field in the snow-atmosphere scenes for off-glint observations. The results are applicable for the inversion of snow and atmospheric data products from the satellite or airborne passive remote sensing measurements above snow. To mitigate the relatively larger differences between measurements and simulation for glint condition as compared to off-glint region, the use of a vertically inhomogeneous snow layer consisting of different ice crystal shapes and sizes is proposed.





This research has been undertaken as part of the investigations in the framework of trans regional (AC)3 project
(Wendisch et al., 2017) that aims to identify and quantify different parameters involved in rapidly changing climate in the
Arctic. In this respect, the analysis of this study will be used to improve the assumptions made for reflectance in snow-
atmosphere system in the algorithms designed to retrieve atmospheric parameters (such as AOT) above Polar Regions.

**Appendix**

For the selected snow models using different ice crystal morphologies, the variation of the snow reflectance $R(\lambda, \Omega)$ at
wavelength $\lambda$ and direction $\Omega$ with respect to the variation $\delta r_e(z)$ of the effective radius profile $r_e(z)$ along the vertical
coordinate z within snow layer can be presented, neglecting nonlinear terms, by the following equation:

$$R(\lambda, \Omega) = R_0(\lambda, \Omega) + \int_0^{Z_t} W_r(z, \lambda, \Omega) \delta r_e(z) dz, \qquad (1)$$

where $R_0(\lambda, \Omega)$ and $R(\lambda, \Omega)$ are the reflection functions calculated assuming an effective radius profile $\acute{r}_e(z)$ and $\acute{r}_e(z) + \delta r_e(z)$
, respectively. The angular variable $\Omega = \{\theta_0, \theta, \varphi\}$ comprises a set of variables: $\theta_0$ is the solar zenith angle, $\theta$ and $\varphi$ are the
zenith and relative azimuthal angles of observation direction; $Z_t$ is the top altitude of snow layer and

$$W_r(Z, \lambda, \Omega) = \frac{\delta R(\lambda, \Omega)}{\delta r_e(z)}, \qquad (2)$$

is the functional derivative of the function $R(\lambda, \Omega)$ with respect to the function $r_e(z)$ which is also called weighting function
(Rozanov et al., 2007). The weighting function was calculated using a numerically efficient forward-adjoint approach
(Rozanov, 2006; Rozanov and Rozanov, 2007) implemented in the SCIATRAN model. Here, it is assumed that properties of
snow do not change in the horizontal plane and within snow layer there is no additional absorber such as soot, dust or other
pollutants. We note that the weighting function includes the contribution of variations not only by the scattering and
extinction coefficients but also by the phase function.

Although the linear relationship given by Eq. (1) can be used to retrieve the vertical profile of the effective radius within
the snow layer in a way similar to that used to the morphology of water droplets (Kokhanovsky and Rozanov, 2012), we
restrict ourselves to the assumption of independent of the altitude $r_e$. Introducing the weighting function for the absolute
variation of the effective radius as:

$$W_r(\lambda, \Omega) = \int_0^{Z_t} W_r(z, \lambda, \Omega) dz, \qquad (3)$$

we have

$$R(\lambda, \Omega) = R_0(\lambda, \Omega) + W_r(z, \lambda, \Omega) \delta r_e. \qquad (4)$$

The resultant linear relationship is a basic equation to formulate inverse problem with respect to the parameter $r_e$ using
measurements of spectral reflectance.

For practical applications Eq. (4) should be rewritten in the vector-matrix form as follows:



$$Y - Y_0 = K(X - X_0). \tag{5}$$

The components of vectors Y and $Y_0$ are the measured and simulated reflectance at discrete number of observation directions $\Omega_j$ and wavelengths $\lambda_i$, the elements of matrix K are weighting functions $W_r(\lambda_i, \Omega_j)$, X=$[r_e]$ is the state vector, $X_0 = [\acute{r}_e]$ is the a priori state vector. We note that in the case under consideration, the matrix K and state vector X are represented by the column vector and scalar, respectively.

Assuming that the number of discrete observation directions $\Omega_j$ and wavelengths $\lambda_i$ is larger than the dimensions of the state vector, the solution of Eq. (5) is obtained by minimizing the following cost function:

$$\Delta = \left\| Y - Y_0 - K(X - X_0) \right\|^2, \tag{6}$$

which describes the root-mean-square deviation between measured and simulated snow reflectance.

Owing to the linear relationship given by Eq. (5) the minimization problem formulated above can be solved analytically:

$$X = X_0 + (K^T K)^{-1} K^T (Y - Y_0). \tag{7}$$

In deriving Eq. (7) we have neglected the linearization error which can be significant if $X_0$ is far from X. To mitigate the impact of linearization error we solve minimization problem given by Eq. (6) iteratively. In particular instead of Eq. (7) is used

$$X_n = X_{n-1} + (K_{n-1}^T K_{n-1})^{-1} K_{n-1}^T (Y - Y_{n-1}), \tag{8}$$

Where n=1, 2, … is the iteration number, $K_{n-1}$ and $Y_{n-1}$ are the matrix of weighting functions and reflectance vector calculated using the state vector $X_{n-1}$. The iteration process is finished if the difference between $X_n$ and $X_{n-1}$ is smaller than a preselected criteria.

The calculation of weighting functions and TOA reflectance is performed at each iteration step using the radiative transfer model SCIATRAN (Rozanov et al., 2014). In SCIATRAN weighting functions are calculated employing a very efficient forward-adjoint technique, which is based on the joint solution of the linearized forward and adjoint radiative transfer equations (Rozanov 2006; Rozanov and Rozanov 2007 and references therein). This enables the TOA reflectance and required weighting function to be calculated simultaneously.

**Acknowledgement**

*We gratefully acknowledge the funding by the Deutsche Forschungsgemeinschaft (DFG, German Research Foundation) – Project Number 268020496 – TRR 172, for SJ within the Transregional Collaborative Research Center "ArctiC Amplification: Climate Relevant Atmospheric and SurfaCe Processes, and Feedback Mechanisms (AC)3". We thank Christine Pohl for her contribution to the extension of ice crystal property database in SCIATRAN.*



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





**Table1.** Summary of the CAR, AERONET aerosol optical thickness (transferred from 0.5 to 0.55 µm) and WFDOAS ozone
data used in this study.

| Dataset number | 1 | 2 | 3 | 4 |
|---|---|---|---|---|
| Date | 7th April 2008 | 7th April 2008 | 7th April 2008 | 15th April 2008 |
| Location | Elson-Lagoon | Elson-Lagoon | Elson-Lagoon | Elson-Lagoon |
| Flight altitude | 206 m | 647 m | 1700 m | 181 m |
| SZA ($\theta_0$) | 70.23° | 69.11° | 67.78° | 62.11° |
| AOT ($\tau\,0.55\,\mu m$) | 0.11 | 0.11 | 0.11 | 0.15 |
| Total ozone column | 416 DU | 416 DU | 416 DU | 463.4 DU |





**Table 2.** Retrieval of physical characteristics of ice crystals with different shape in the case of most roughened habits. Underlined numbers indicate minimum RMSE.

| Ice crystal habit | Retrieved effective radius (μm) | | Old snow | | Fresh snow | |
|---|---|---|---|---|---|---|
| | Old snow | Fresh snow | Bias (%) | RMSE (%) | Bias (%) | RMSE (%) |
| Fractal | 69.37 | 76.06 | 3.50 | 9.75 | 13.16 | 14.69 |
| Droxtal | 94.48 | 106.95 | 0.87 | 25.54 | 10.10 | 34.14 |
| Column | 74.71 | 80.49 | 2.17 | 7.32 | 12.36 | 15.72 |
| Hollow column | 67.32 | 72.85 | 2.80 | 11.15 | 13.66 | 15.14 |
| Aggregate of 8 columns | 98.83 | 107.62 | 2.79 | 6.97 | 11.85 | 18.27 |
| Plate | 38.93 | 61.44 | -0.44 | 21.47 | 11.68 | 16.99 |
| Aggregate of 5 plates | 78.02 | 83.41 | 1.82 | 10.34 | 11.23 | 12.85 |
| Aggregate of 10 plates | 65.36 | 69.28 | 2.34 | 13.91 | 11.52 | 13.16 |
| Hollow-bullet rosette | 67.01 | 73.28 | 2.16 | 9.99 | 12.71 | 15.16 |

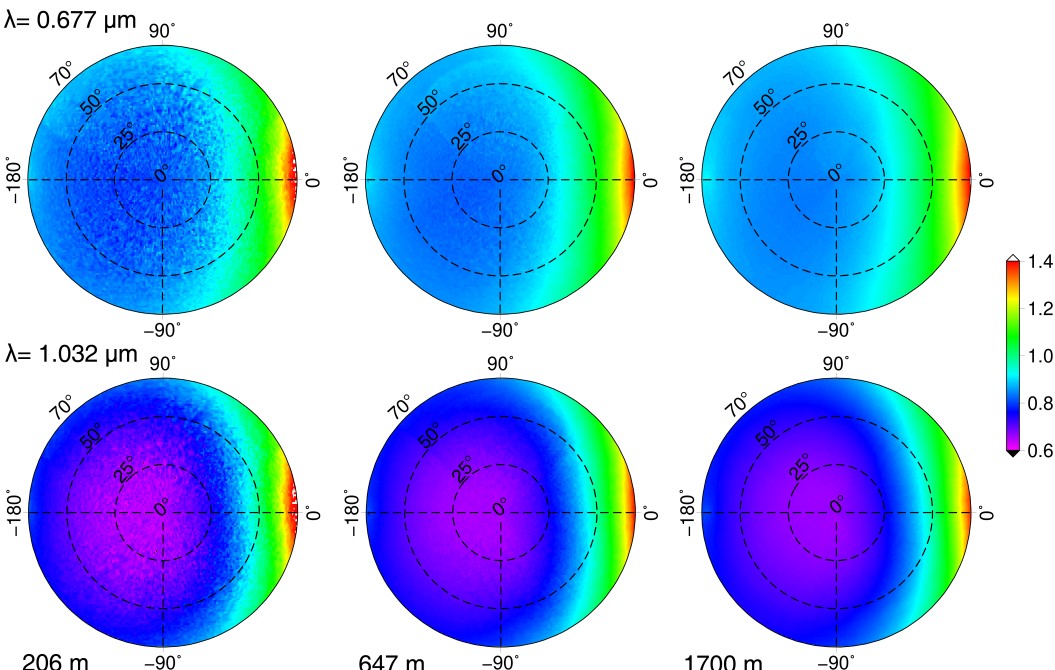

**Figure 1:** Angular distribution of reflectance in the snow-atmosphere system derived from CAR measurements on 7[th] of April 2008, at Elson Lagoon (71.3° N, 156.4° W): Upper panel at 0.677 µm wavelength and 3 flight altitudes: 206, 647 and 1700 m, respectively; lower panel at 1.032 µm wavelength and at the same flight altitudes. The principal plane is the horizontal line ($\varphi = 0°$ and 180°), viewing zenith angle is shown as the radius of polar plots from 0° (nadir) to 70°.

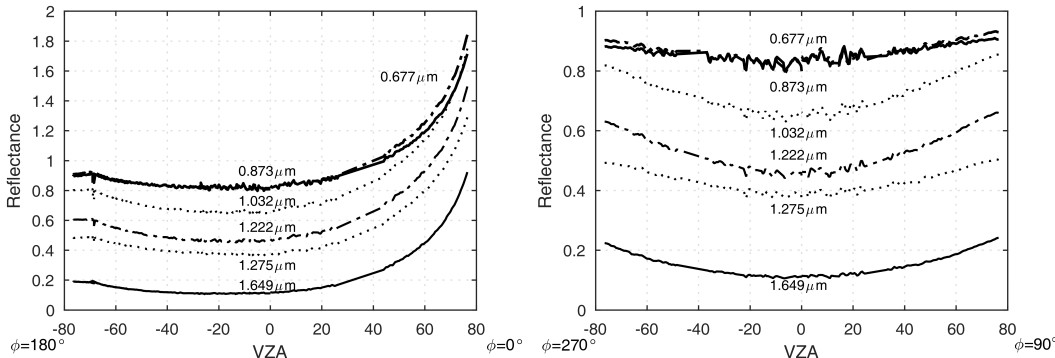

**Figure 2:** Angular distribution of reflectance in the snow-atmosphere system, derived from measurements by CAR at 647 m flight altitude and six wavelengths: 0.677 µm, 0.873 µm, 1.032 µm, 1.222 µm, 1.275 µm and 1.649 µm, on 7[th] of April 2008, at Elson Lagoon (71.3° N, 156.4° W); left panel: in the principal plane (φ = 0° and 180°) and right: cross plane (φ = 90° and 270°).





**Figure 3:** The change of reflectance values in principal plane (φ = 0° and 180°) with size and shape of ice crystals at the wavelength of 1.649 μm.



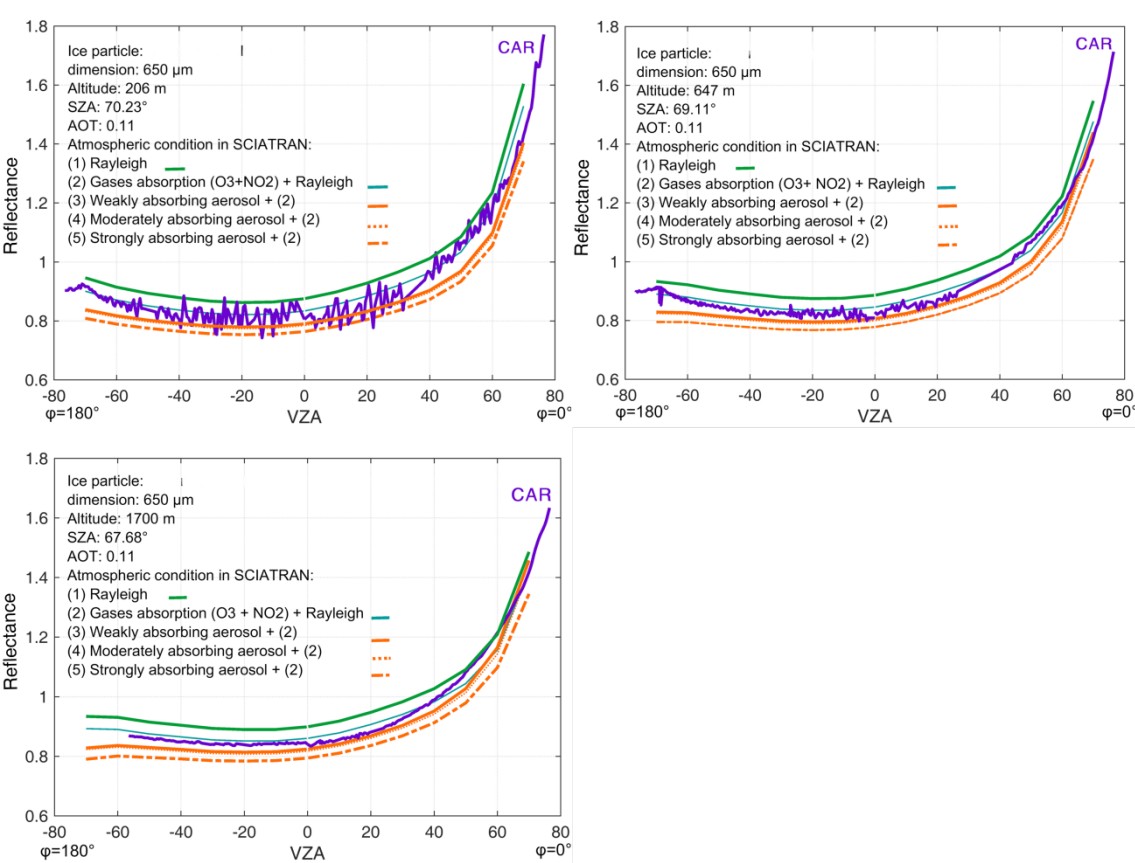

**Figure 4:** Measured and simulated reflectance at 0.677 μm versus VZA in the principal plane (φ = 0° and 180°) at three different flight altitudes. Upper left, upper right and the lower left panel represent results at 206, 647 and 1700 m flight altitude respectively. The green lines indicate simulated reflectance assuming Rayleigh scattering (case i); the blue line shows reflectance for case ii (as case i including absorption of $O_3$ and $NO_2$, the orange lines show the reflectance for case iii (as case ii but adding aerosol with an AOT of 0.11 for three types of aerosol: i) weakly absorbing, ii) moderately absorbing and iii) strongly absorbing).





**Figure 5:** Comparison of measured and simulated reflectance. Measurements (shown by triangles) were performed by the CAR instrument over old snow at 647 m flight altitude on the 7[th] of April 2008 at 1.649 μm. SCIATRAN simulations in the principal and cross plane given by the dashed-dotted and dotted lines respectively by different colors: green, blue and red present smooth, moderately roughened and severely roughened crystal surface. Positive and negative VZAs correspond to azimuthal angles φ = 0° and 180° for principal plane and φ = 90° and 270° for perpendicular plane respectively.






**Figure 6:** The same as Fig. 5 but the measurements by the CAR instrument were performed on the 15th April at 181 m flight altitude over fresh snow.



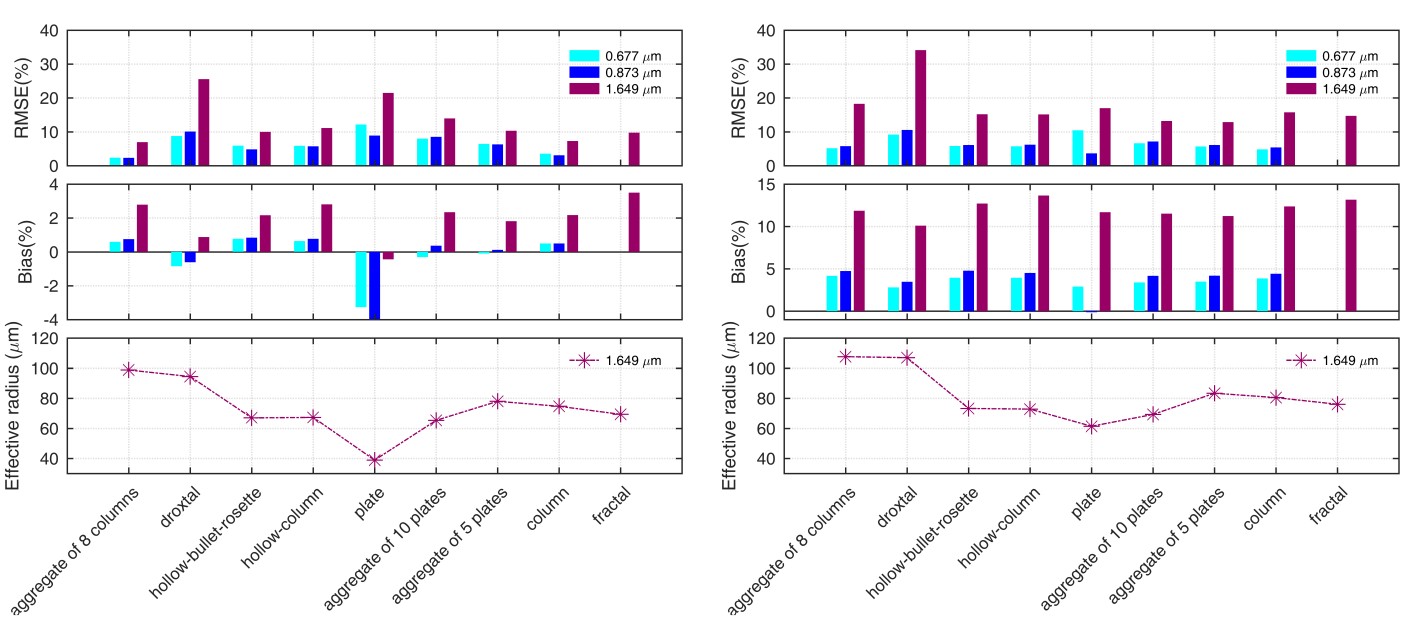

**Figure 7:** Comparison of snow grain size retrieval and best fit of reflectance at three wavelengths: 0.677, 0.873 and 1.649 μm, left panel: old snow case and right panel: fresh snow case.





**Figure 8:** Left column shows reflectance at three wavelengths: 0.677, 0.873 and 1.649 µm from the CAR measurements acquired on 7$^{th}$ of April 2008, at Barrow/Utqiaġvik Alaska at an altitude of 647 m; The middle column depicts the absolute difference between simulation and measurement: (RSCIATRAN – RCAR); The right column shows the relative difference in (%).



λ= 0.677 µm

λ= 0.873 µm

λ= 1.649 µm

**Figure 9:** The same as Fig. 8 but the measurements by CAR instrument were performed on 15ᵗʰ April 2008 at 181 m flight altitude over fresh snow.





**Figure 10:** The scatter plot with corresponding pearson correlation coefficient of reflectance measured by CAR and simulated by SCIATRAN; here the color bar represents number density of pixels.