# Peer review of "Simulated reflectance above snow, constrained by airborne measurements of solar radiation: Implications for the snow grain morphology in the Arctic"

_Atmospheric Measurement Techniques, 2020_

## Short Comment (SC1) · 17 Apr 2020

I support the publication of this paper. One major comment is that the selection of the appropriate snow grain shape and size must be performed usnig both angular and spectral measurements. The authors discuss mainly the angular patterns. It is interesting to see how differ spectral reflectances for different best models shown in Table 2 and how they agree with spectral reflectance measurements at 14 Cloud Absorption Radiometer (CAR) spectral channels. Please, list the CAR channels in the paper. The authors can easily reproduce such a figure using SCIATRAN. Also asymmetry param-

eters in the visible must be given for all cases shown in Table 2. The authors assume clean snow. I think, the authors must show some evidence in the paper that the measured spectra have not been affected by possible snow pollution. Minor comments:line 4, leads–>lead;line 32, to be –>is;line 46 (AA);line 204, is reference available?;line 210, did you assume rough Koch crystals?;line 240, the wavelength of 1.24 microns is more suitable for the grain size retrieval ( larger sensitivity to the grain size);line 250, matrix->function;line 271, a priori;line 276, remove 'on the snow layer';line 295, remove #please#.

---

## Referee Comment (RC1) · Alexander Kokhanovsky (Referee) · 24 Apr 2020

I support the publication of this paper. One major comment is that the selection of the appropriate snow grain shape and size must be performed usnig both angular and spectral measurements. The authors discuss mainly the angular patterns. It is interesting to see how differ spectral reflectances for different best models shown in Table 2 and how they agree with spectral reflectance measurements at 14 Cloud Absorption Radiometer (CAR) spectral channels. Please, list the CAR channels in the paper. The authors can easily reproduce such a figure using SCIA-

[Figure]

TRAN. Also asymmetry parameters in the visible must be given for all cases shown in Table 2. The authors assume clean snow. I think, the authors must show some evidence in the paper that the measured spectra have not been affected by possible snow pollution. Minor comments:line 4, leads–>lead;line 32, to be –>is;line 46 (AA);line 204, is reference available?;line 210, did you assume rough Koch crystals?;line 240, the wavelength of 1.24 microns is more suitable for the grain size retrieval ( larger sensitivity to the grain size);line 250, matrix->function;line271, apriori;line276,remove'onthesnowlayer';line295, remove #please#.

---

## Referee Comment (RC2) · Anonymous Referee #2 · 25 Aug 2020

Within this manuscript, simulations of the bidirectional reflectance factor (BRF) are compared to airborne measurements of the BRF of snow surfaces with the Cloud Absorption Radiometer (CAR) at three different wavelengths in the visible and near infrared wavelength range. The measurements were part of the ARCTAS spring campaign over Alaska in 2008. In order to simulate the BRF correctly with the radiative transfer model SCIATRAN, many input parameters such as snow (e.g., snow grain size and ice crystal shape) and atmospheric parameters (e.g. aerosol optical thickness, ozone concentration) need to be measured or independently retrieved. Subsequently,

the BRF simulations utilizing the carefully selected input parameters are compared to the CAR measurements, showing in general a good agreement. In comparing radiative transfer simulations of the combined snow-atmosphere system with actual measurements, this study presents an important contribution that is potentially relevant for passive remote sensing above snow surfaces. The authors efforts to carefully select the input parameters and presenting a retrieval algorithm for ice crystal shape and snow grain size from reflectance measurements should be acknowledged.

The figures are generally of good quality, which helps to convey the arguments of the authors. However, there are some aspects that need further focus in my opinion, After some general comments, the more specific comments and suggestions for technical corrections follow below.

**General comments**

I have a couple of general comments that need further attention: (1) about the scope of the manuscript, (2) about the reflectance terminology used throughout the manuscript, (3) the use of English, (4) the handling of measurement uncertainties, (5) the definition of the snow grain size.

(1) After reading the manuscript, its scope is still not entirely clear to me: (1) Is it about the description of a novel algorithm for the simultaneous retrieval of the snow grain size and ice crystal shape? In that case, it is a bit confusing to me that you moved the entire description of this algorithm to the appendix, while at the same time, the sentence 'we present a novel two-stage snow grain morphology [...] retrieval algorithm' is part of the abstract. If you want to focus more on the description of this algorithm, maybe it is worth to think about moving it to a more prominent spot within the manuscript. (2) Is it about the sensitivity of the radiative transfer model to the snow and atmospheric input parameters? (3) Or is the main focus the comparison of BRF simulations with the CAR measurements? To be clear, I do think that all three parts are important contributions. However, it is important that each part presented in the manuscript is investigated

AMTD
thoroughly. And until now, each part is missing some pieces in my point of view and I will give more details on that further below in the specific comments. However, this lack of focus seems to already appear in the title, which reads very confusing and imprecise to me. The authors of course wanted to include all pieces, but this came at the cost of the readability and conciseness.

(2) Another important part is the terminology used throughout the manuscript. As you are referencing Schaepman-Strub et al. (2006) extensively in Sect. 2 'Theoretical background', I recommend you also stay consistent in the use of reflectance terminology. Equation 4 defines a reflectance factor according to Schaepman-Strub et al. (2006) and should be named 'reflectance factor' and not 'reflectance' as stated for example on Page5 Line143. Otherwise, this quickly becomes very confusing to the reader as it is very important to stay precise to differentiate between the different reflectance quantities. I mention some occasions where 'reflectance' should be replaced with 'reflectance factor' below in the Technical corrections. However, the authors should double check and change it in the entire manuscript.

(3) Unfortunately, some parts of the manuscript are quite difficult to read and the use of English should be improved to make the line of argumentation easier to follow. I gave some recommendations in the comments, but I think the authors should check the entire manuscript to foster reading comprehension.

(4) One of the most pressing aspects is the lack of accounting for measurement uncertainties. A detailed discussion of the measurement and retrieval uncertainties for the BRF measurements with the CAR is missing. Also, every time CAR measurements are shown, uncertainty bars need to be included (especially Figures 5 and 6, see also specific comments below). This also applies to the a priori estimation of the effective radius of the snow grains (Figure 7), and the scatter plots in Figure 10. I understand that adding uncertainty bars for the simulations in Figure 3 is not applicable as the plot is already very busy. However, as you even test the sensitivity of the simulations with respect to, e.g. how absorbing the aerosols are, at least some uncertainty estimates Interactive comment

should be given within the text. The uncertainty discussion is especially important as it might influence conclusions drawn from the RMSE analysis: if all influencing factors are named and properly quantified, non-significant differences in the RMSE of 0.4 % (Figure 5) between two different ice crystal shapes should not be relevant and influence a decision for a specific ice crystal shape being used in the simulations.

(5) The snow grain size is given in terms of the maximum extent within the manuscript. Although the physical size of a snow grain is traditionally defined by the length of the largest extension of the crystal, in terms of radiative properties the optical-equivalent snow grain size is way more important. It is defined as the radius of a collection of spheres with the same total volume and surface area compared to the actual non-spherical snow grain (see e.g. Grenfell and Warren, 1999; Neshyba et al., 2003). Displaying the reflectance factor for different crystal shapes and sizes in Figure 3, one could assume that the same crystal sizes are comparable between the different shapes. However, from a radiative point of view, this is not true, as each size (largest extent in your case) is defined differently. I recommend using an optical-equivalent snow grain size in Figure 3 instead of the largest extension.

Grenfell, T. C. and Warren, S. G.: Representation of a nonspherical ice particle by a collection of independent spheres for scattering and absorption of radiation, J. Geophys. Res., 104, 31.697–31.709, doi:10.1029/1999JD900496, 1999.

Neshyba, S. P., Grenfell, T. C., and Warren, S. G.: Representation of a nonspherical ice particle by a collection of independent spheres for scattering and absorption of radiation: 2. Hexagonal columns and plates, J. Geophys. Res., 108, Art. No. 4448, doi:10.1029/2002JD003302, 2003.

Specific comments

Abstract: (1) The first 1.5 paragraphs (L10-L19) are too general for an abstract. SCIA-TRAN is the first really specific information about the study presented in this manuscript on Line 20. Please try to include specific information already earlier and leave some Interactive comment

of the general motivation to the section 'Introduction'. (2) L27: specify the used wavelength channels at this point. (3) L31: round the effective radius to an integer number as the two decimals imply a precision which is not achievable.

P2L41: please add the Arrhenius reference

P5L129: It is very important to list the different atmospheric contributions to the measured radiance. However, please be a bit more precise in the formulation: for example, the scattering by the atmosphere before and after reaching the surface is not removed. More precisely, 'the contribution of light scattered by the atmosphere both before and after being reflected from the surface' is removed (see Schaepman-Strub et al., 2006). Please specify the four contributions accordingly, referring to the different contributions of scattered radiation reaching the instrument's field of view.

P5L134: please already give the CAR wavelength range at this point.

P5L138-149: This paragraph is very important to understand the quantities measured and simulated within this study. However, it is currently difficult to read. I recommend to reformulate the sentences and taking special care with regard to the sentence structure. This comment includes for example: P5L139: is applied to the measured radiances; P5L143: In the simulation [...]: this sentence is unclear, please reformulate. P5L147: We assume that the reflectance factor at flight altitude is a good approximation of the BRF just above the surface at infrared wavelengths where atmospheric scattering is negligible; Eq. 4: the subscript '0' should be defined at this point.

Sect. 3: Subheadings would improve the readability considerably. I recommend to start with some more details about the ARCTAS spring campaign, adding a map with the flight tracks of the measurements used in this study, before giving details about the CAR instrument and the ozone and nitrogen dioxide data.

Figure 1: please add the position of the Sun in the caption of the figure to make it immediately clear where the forward and backward scattering directions are.

AMTD
Figure 2: missing whitespace before 1.649 um in the figure caption. The y axis should be named 'Reflectance factor', as this is what you calculate from Eq. 4. Both Figure 1 and 2 should be described in more detail and not only mentioned in the text.

P6L173: You are giving an explanation for the decrease in inhomogeneities in the BRDF data. Please also discuss the increase of the BRF with altitude.

P6L182: In the paragraph describing the AOT data, a quick description of the representativeness of the aerosol conditions during the ARCTAS spring campaign with respect to the Barrow climatology would be helpful.

P6L183: please provide some more details about the spaceborne measurements of total column ozone.

P7L200: I guess the measurement location is sufficiently remote to justify this assumption. However, are there any measurements of black carbon on snow available for this region to further provide evidence for this?

P8L229: please provide more details about the 'exponential vertical distribution' used for the vertical profile of the aerosol number density. Are you assuming the aerosol number density is reduced exponentially with height? Is this not influenced by the boundary layer height? And why were 3 km chosen when the measurements were conducted at flight altitudes below 1700 m? Also: for the vertical profiles of pressure and temperature, did you use monthly mean profiles as well or could you make use of radiosonde launches in the vicinity of the study area?

Figure 3: (1) The ice crystal shapes presented in Figure 3 do not match the 9 morphologies introduced on P7L206: it seems you are presenting solid bullet rosettes in the figure, which are not mentioned in the text. On the other hand, you are not presenting the results for the fractal particles. Please clarify that as it is a bit confusing to me. (2) I assume this is still the calculated Reflectance factor, please name the y axis accordingly. AMTD
P9L264: please specify 'same size', as in the sentence before you are talking about a size range between 60 to 10000 um.

P9L266: please clarify this sentence, because when I look at Figure 3, also for the plate shape the reflectance factor increases in the backward direction compared to the nadir direction.

P9L267: larger reflectance in all directions compared to what? The reflectance factor for the hollow bullet rosette seems to be at least equally high for some snow grain sizes compared to the aggregates of 5 or 10 plates.

Figure 4: (1) the green and blue lines and too similar and are hard to distinguish within the plot. (2) this is a reflectance factor again? Please name the y axis accordingly.

Figures 5 and 6: (1) this is a reflectance factor again? Please name the y axis accordingly. (2) The uncertainty of the CAR measurements needs to be included in the figure in the form of error bars. This also needs to be considered when calculating the RMSE. I assume a difference in RMSE of less than 0.4 % as visible between the chosen aggregates of 8 columns (98.8 um) and the columns (74.7 um) is not significant when considering possible measurement and retrieval uncertainties. This needs to be discussed in the manuscript. (3) The surface roughness clearly affects the CAR measurements at large viewing zenith angles. As I understand, the macroscopic surface roughness (in contrast to the ice crystal roughness) is not included in the SCIATRAN simulations? In this case, I suspect you are trying to fit the simulations to the measurements using different single scattering properties for the different ice crystal shapes, while more probably the macroscopic surface roughness is the underlying reason for the deviations between CAR measurements and SCIATRAN simulations. Macroscopic surface roughness enhances the backscatter by changing the effective angle of incidence, and reduces the forward scatter by casting shadows. Of course, this depends on the size of the roughness structures and their orientation, and I guess both parameters are unknown for the measurement conditions? Maybe some observations from
within the aircraft with the naked eye or camera pictures could give an indication? At least the reduction in forward scattering of the CAR measurements compared to the simulations is visible for many different ice crystal shapes in Figure 5. Figure 6, however, shows an increase in the forward scattering as measured with CAR. In trying to choose the lowest RMSE for model simulations that neglect macroscopic surface roughness, it seems to me you can partly mimic the effect of surface roughness in choosing different ice crystal shapes (and single scattering properties). Thus, you are getting the 'right simulation', but for the wrong reasons in my point of view. Is there any way to test your simulations for different macroscopic surface roughness heights and orientation? Either way, this uncertainty needs to be discussed in detail within the manuscript.

P11L333-345: The justification of the ice crystal shape retrievals with the temperaturedependence seems dubious to me. One needs to be careful in differentiating the important temperatures here. It is true that temperature (and supersaturation!) strongly affect the shape of pristine ice crystals when the precipitating snow is formed within the cloud. If anything, the ice crystal shape should be connected to the temperature profile at the time of the last snowfall (excluding snow aging processes). However, the temperatures you are stating are temperatures measured in-flight, probably days after the precipitation event. This temperature is completely unrelated to the snow on the ground, especially as you report yourself that the snow surface consists of old snow during most days. After the snow has fallen to the ground, the vertical temperature gradient at the surface and within the snowpack is way more important for the ice crystal shape (influencing snow metamorphism processes). If you don't have in situ observations looking at the ice crystal shape on the ground, you cannot validate your ice crystal shape retrieval in that way.

Figure 7: This is way more illustrative and provides more information than Table 2, which becomes redundant in my point of view and can be removed from the manuscript.

**AMTD**
P12L359: it seems you are normalizing the RMSE somehow. Please provide the formula how you calculated the RMSE, as your description in the text seems to be imprecise.

P12L364: please round the effective radii to integer values. Providing two decimals is implying a degree of accuracy which is not achieved.

Figure 8: caption: 'reflectance factor'

P12L367: I would recommend introducing Figure 10 only after Figures 8 and 9.

Figure 10: (1) caption: 'reflectance factor', x and y axis: 'reflectance factor', please state again in the caption which columns belong to the old and new snow cases. (2) I am interested in seeing a comparison of the correlation coefficients between new snow case and the lowest flight level of the old snow case as they have roughly comparable flight altitudes. This might make it easier to discuss a possible influence of surface inhomogeneities. At this point it would also help to provide more details about the differences in flight tracks between the two measurement days. Was the same area probed on both days? Otherwise of course, even the same flight altitude might not be comparable. This is connected to my earlier comment to provide more details about the actual flights performed during the campaign.

P12L377: I do not agree with the conclusion drawn here. The high correlation coefficient and small discrepancies do not justify the selection of this wavelength channel for the selection of the best ice crystal shape. The correlation coefficient and small bias is made 'by construction', as you selected the ice crystal shape based on the lowest bias between simulation and CAR measurements in the first place. The high correlation coefficients for this wavelength channel are therefore not surprising. In addition, the last sentence of this section (P12L379) seems a bit out of place and should be rephrased.

P13L388: This is an important point and should be included in this study already by looking at the correlation coefficient between measured and simulated reflectance fac-

AMTD
tors and their dependence on the flight altitude for the case of old snow. I am interested to see whether there is a clear dependence of the correlation coefficient on the flight altitude.

P13L405: With regard to my earlier comment, the justification of the ice crystal shape retrieval with the temperature dependence cannot be mentioned here (and also not in the abstract).

P13L416: I wonder why the use of a vertically inhomogeneous snow layer in the model is only mentioned here and not in the discussion of the results already. It should not be mentioned for the first time in the Conclusions in my point of view.

Technical corrections

P1L30: Assuming that the snow layer consists [...]

P3L78: delete ';' after '2011'

P3L88: 'phenomenological', 'airborne'

P5L126: of the surface

- P5L127: on the measured radiance
- P5L128: scattering or absorption applying RTMs
- P5L128: This removes the four atmospheric [...] from the measured radiance: i) [...]
- P5L153: delete 'etc' or be more specific

P6L160: by a mirror – missing blank

P6L163: do you mean viewing zenith and azimuth angles?

P6L163: Please rephrase: The high  $[\ldots]$  resolution  $[\ldots]$  allows the estimation of the anisotropy of the reflectance in the snow-atmosphere system with high accuracy.

P6L167: RTM simulations
P6L175: do you mean spatial inhomogeneity? P7L208: eight ice crystal shapes/habits P7L208: (referred to as fractal in this paper) P8L227: ground-based measurements from AERONET P8L228: selecting one of the aerosol types P9L271: a priori knowledge P9L272: to accurately reproduce measurements P9L279: evaluate the impact of the atmosphere P9L282: assuming the following properties P12L362: described in the previous sections P12L380: measurement of the reflectance factor P13L390: reflectance factor P13L397: reflectance factor P13L408: reflectance factor P13L411: reflectance factor P14L429: comma misplaced P15L466: do you really mean TOA reflectance? Or reflectance at flight altitude?

**AMTD**

---

## Author Comment (AC1) · 21 Sep 2020

The comment was uploaded in the form of a supplement:
https://amt.copernicus.org/preprints/amt-2020-58/amt-2020-58-AC1-supplement.pdf

---

## Author Comment (AC2) · 21 Sep 2020

**Reply to Referee #2**

We would like to thank you for the review and your constructive comments which helped to improve our manuscript.

Our point-by-point responses to the specific comments (in red) are given in blue and the modification made in the manuscript is presented in green. This document also includes a marked-up version of manuscript..

Best Regards,

Soheila Jafariserajehlou

**General comments**

**#1 Comment to the Author**

After reading the manuscript, its scope is still not entirely clear to me: (1) Is it about the description of a novel algorithm for the simultaneous retrieval of the snow grain size and ice crystal shape? In that case, it is a bit confusing to me that you moved the entire description of this algorithm to the appendix, while at the same time, the sentence 'we present a novel two-stage snow grain morphology […] retrieval algorithm' is part of the abstract. If you want to focus more on the description of this algorithm, maybe it is worth to think about moving it to a more prominent spot within the manuscript. (2) Is it about the sensitivity of the radiative transfer model to the snow and atmospheric input parameters? (3) Or is the main focus the comparison of BRF simulations with the CAR measurements?

To be clear, I do think that all three parts are important contributions. However, it is important that each part presented in the manuscript is investigated thoroughly. And until now, each part is missing some pieces in my point of view and I will give more details on that further below in the specific comments. However, this lack of focus seems to already appear in the title, which reads very confusing and imprecise to me. The authors of course wanted to include all pieces, but this came at the cost of the readability and conciseness.

**Author's response:**

We agree with your comment, we need to highlight more the main focus of our paper. As you said all three parts are very important, we tried to bring necessary information and avoid of representing too much details. But we do understand your point and therefore, we modified the abstract and the title of our paper.

We would like to emphasize that the goal of our study was to achieve the best possible simulated reflectance in a snow-atmosphere system using our radiative transfer model. Testing the sensitivity of reflectance to snow and atmospheric parameters was a step to show the importance of snow morphology and atmosphere.

To achieve our goal and minimize the difference between simulation and observation, we tried to have the best possible estimation about surface and atmosphere. Therefore, the use of snow grain size/shape retrieval algorithm was our second priority and for this reason we put the algorithm itself in the appendix to keep the structure of manuscript in the frame of main goal.

We agree that the sentence "we present a novel two stage algorithm in the abstract", and the title of our paper may be confusing. To solve this problem we applied the following changes.

**Modifications:**

Title: Simulated reflectance above snow, constrained by airborne measurements of solar radiation: Implications for the snow grain morphology in the Arctic.

Line 7-10:

In this paper, we simulate the reflectance in a snow-atmosphere system using the phenomenological radiative transfer model SCIATRAN and compare the results with that of airborne measurements. To minimize the differences between measurements and simulation, we determine and employ the key atmospheric and surface parameters such as snow morphologies (or habits)…
* * *
**#2 Comment to the Author**

Another important part is the terminology used throughout the manuscript. As you are referencing Schaepman-Strub et al. (2006) extensively in Sect. 2 'Theoretical background', I recommend you also stay consistent in the use of reflectance terminology. Equation 4 defines a reflectance factor according to Schaepman-Strub et al. (2006) and should be named 'reflectance factor' and not 'reflectance' as stated for example on Page5 Line143. Otherwise, this quickly becomes very confusing to the reader as it is very important to stay precise to differentiate between the different reflectance quantities. I mention some occasions where 'reflectance' should be replaced with 'reflectance factor' below in the Technical corrections. However, the authors should double check and change it in the entire manuscript.

**Author's response:**

Thanks for pointing this out. Yes, we should stay consistent with respect to terminology. We changed reflectance to "reflectance factor" in every place (text and figures) where we refer to calculated/simulated reflectance factor using Eq. 4.

**Modifications:**

Please see the manuscript.
* * *
**#3 Comment to the Author**

Unfortunately, some parts of the manuscript are quite difficult to read and the use of English should be improved to make the line of argumentation easier to follow. I gave some recommendations in the comments, but I think the authors should check the entire manuscript to foster reading comprehension.

**Author's response:**

We applied all of your comments and we also tried to improve the readability of our manuscript.

**Modifications:**

Please see the technical comments with details and manuscript.
* * *
**#4 Comment to the Author**

One of the most pressing aspects is the lack of accounting for measurement uncertainties. A detailed discussion of the measurement and retrieval uncertainties for the BRF measurements with the CAR is missing. Also, every time CAR measurements are shown, uncertainty bars need to be included (especially Figures 5 and 6, see also specific comments below). This also applies to the a priori estimation of the effective radius of the snow grains (Figure 7), and the scatter plots in Figure 10. I understand that adding uncertainty bars for the simulations in Figure 3 is not applicable as the plot is already very busy. However, as you even test the sensitivity of the simulations with respect to, e.g. how absorbing the aerosols are, at least some uncertainty estimates should be given within the text. The uncertainty discussion is especially important as it might influence conclusions drawn from the RMSE analysis: if all influencing factors are named and properly quantified, non-significant differences in the RMSE of 0.4 % (Figure 5) between two different ice crystal shapes should not be relevant and influence a decision for a specific ice crystal shape being used in the simulations.

**Author's response:**

We agree that uncertainty of measurements, simulation and retrieval are very important and should be plotted and discussed in our manuscript. Thanks for pointing this out.

The underlined of CAR measurements is within 5% based on a comprehensive study done in NASA in 2007 (Dr. Gatebe from NASA / co-author of this paper). We added uncertainty envelopes to measurements in Fig. 5, 6 (now they are 6 and 7) and also in the text.

The uncertainty of effective radius retrieval is estimated to be ~10% on the base of optimal estimation technique. We also added uncertainty envelope to the plot in Fig.7 (now is 8) and text accordingly.

The uncertainty of our radiative transfer calculations is estimated to be in the range of 0.1 % and we did not add it to our plots because of being too small, because it can't be seen. However, we added this uncertainty to the text of manuscript.

**Modifications:**

Line 342 – 344:

Fig. 6 shows one example of the comparison between measured and simulated reflectance factor at principal and cross planes. The absolute uncertainty of CAR measurements is within 5% and shown by uncertainty envelope. The accuracy of our radiative transfer calculations is estimated to be in the range of 0.1 %.

Line 385-386:

The uncertainty of effective radius retrieval is estimated to be ~10% on the base of optimal estimation technique and shown by gray envelope in Fig. 8.

[Figure]

[Figure]

[Figure]
* * *
**#5 Comment to the Author**

The snow grain size is given in terms of the maximum extent within the manuscript. Although the physical size of a snow grain is traditionally defined by the length of the largest extension of the crystal, in terms of radiative properties the optical-equivalent snow grain size is way more important. It is defined as the radius of a collection of spheres with the same total volume and surface area compared to the actual nonspherical snow grain (see e.g. Grenfell and Warren, 1999; Neshyba et al., 2003). Displaying the reflectance factor for different crystal shapes and sizes in Figure 3, one could assume that the same crystal sizes are comparable between the different shapes. However, from a radiative point of view, this is not true, as each size (largest extent in your case) is defined differently. I recommend using an optical-equivalent snow grain size in Figure 3 instead of the largest extension.

**Author's response:**

Yes, for this figure and relevant explanation we used the maximum length (but not for the retrieval of snow grain size). We agree that the maximum length is not giving the full picture. We thought maximum length would give a better/easier imagination of grain sizes to the readers when comparing different shape. But we understand your point and so added the effective radius besides maximum length to this figure (Fig. 4). We explained this new column in the caption and gave its definition within the manuscript.

[Figure]

**Specific comments:**

**#1 Comment to the Author**

Abstract: (1) The first 1.5 paragraphs (L10-L19) are too general for an abstract. SCIATRAN is the first really specific information about the study presented in this manuscript on Line 20. Please try to include specific information already earlier and leave some of the general motivation to the section 'Introduction'.

**Author's response:**

We modified the abstract and moved most of the first paragraph to introduction. We also brought key information about SCIATRAN to the beginning of abstract.

**Modifications:**

Line 1-10:

Accurate knowledge of the reflectance from snow/ice covered surface is of fundamental importance for the retrieval of snow parameters and atmospheric constituents from space-based and airborne observations. In this paper, we simulate the reflectance in a snow-atmosphere system using the phenomenological radiative transfer model SCIATRAN and compare the results with that of airborne measurements. To minimize the differences between measurements and simulation, we determine and employ the key atmospheric and surface parameters such as snow morphologies (or habits).
* * *
**#2 Comment to the Author**

L27: specify the used wavelength channels at this point.

**Author's response:**

Done.

**Modifications:**

Line 22-23:

…with that from airborne CAR measurements in the visible (0.670 μm) and NIR (0.870 and 1.6 μm) wavelength range.
* * *
**#3 Comment to the Author**

L31: round the effective radius to an integer number as the two decimals imply a precision which is not achievable.

**Author's response:**

Yes we agree, we rounded the numbers.

**Modifications:**

Line 26: ...an effective radius ~ 99

line 28: ...an effective radius ~ 83
* * *
**#4 Comment to the Author**

P2L41: please add the Arrhenius reference

**Author's response:**

Sorry we forgot it. We added the reference.

**Modifications:**

Line 39 and line 522:

Arrhenius, S., On the influence of carbonic acid in the air upon the temperature of the ground. Mag. J. Sci., London, Edinburgh, Dublin Phil, 41, 237-276, 1896.
* * *
**#5 Comment to the Author**

P5L129: It is very important to list the different atmospheric contributions to the measured radiance. However, please be a bit more precise in the formulation: for example, the scattering by the atmosphere before and after reaching the surface is not removed. More precisely, 'the contribution of light scattered by the atmosphere both before and after being reflected from the surface' is removed (see Schaepman-Strub et al., 2006). Please specify the four contributions accordingly, referring to the different contributions of scattered radiation reaching the instrument's field of view.

**Author's response:**

We modified the text accordingly.

**Modifications**: line 129-134

This removes the four atmospheric contributions from the measured radiance at TOA or flight altitude (Schaepman-Strub et al., 2006): the contribution of light scattered by the atmosphere: i) before the solar radiation has reached the surface, ii)after being reflected by the surface, iii) before and after reaching the surface and iv) the atmospheric path radiance.
* * *
**#6 Comment to the Author**

P5L134: please already give the CAR wavelength range at this point.

**Author's response:**

We added the wavelength range.

**Modifications:** line 137-138:

Sensitivity studies have demonstrated that atmospheric contributions to the CAR channel observations range from 3 to 12% depending on wavelength in the range of 0.381 to 2.324 μm.
* * *
**#7 Comment to the Author**

P5L138-149: This paragraph is very important to understand the quantities measured and simulated within this study. However, it is currently difficult to read. I recommend to reformulate the sentences and taking special care with regard to the sentence structure. This comment includes for example: P5L139: is applied to the measured radiances;

**Author's response:**

We corrected all paragraphs here and reformulated the sentences to transfer the message better.

**Modifications**: line 141-157

The atmospheric correction methods relies on different assumptions by which several source of uncertainties should be taken into account. In this study, to avoid such uncertainties, we do not apply an atmospheric correction to the measurements (radiances $L_{r,h}$) at flight altitude (h).. Instead, we calculate and use the reflectance at flight altitude by the following equation:
* * *
$$R = \frac{\pi L_{r,h}\left(\theta_i, \theta_r, \Delta\varphi\right)}{F_{0,\lambda}\cos\theta_i} \tag{4}$$

where $L_{r,h}$ is the measured radiance at flight altitude. All reflectance/$BRF_\lambda^e$ values at flight altitude in this study represent  R in Eq. 4 and are referred to as "reflectance factor" in the snow-atmosphere system.

In the simulation of the reflectance factor in a coupled snow-atmosphere system, we need to account for atmospheric effects contribution properly. For this reason, we take independent data about atmospheric parameters (Aerosol Optical Thickness (AOT) and gases absorption) from ground-based and space-borne measurements. We select the data with the closest spatial and temporal interval  actual airborne measurements. .

We discuss more details of the atmospheric data and their application to the simulation routine in sect. 3 and 4. To estimate $BRF_\lambda^e$ just above the surface, further atmospheric correction is needed. We assume the reflectance factor at flight altitude is a good estimation of $BRF_\lambda^e$ just above the surface at infrared wavelengths where atmospheric scattering is negligible.
* * *
**#8 Comment to the Author**

P5L143: In the simulation [...]: this sentence is unclear, please reformulate.

**Author's response:**

Done.

**Modifications**: Line 149-151:

In the simulation of the reflectance factor in a coupled snow-atmosphere system, we need to account for atmospheric effects contribution properly. For this reason, we take independent data about atmospheric parameters (Aerosol Optical Thickness (AOT) and gases absorption) from ground-based and space-borne measurements.
* * *
**#9 Comment to the Author**

P5L147: We assume that the reflectance factor at flight altitude is a good approximation of the BRF just above the surface at infrared wavelengths where atmospheric scattering is negligible;

Author's response:

We applied the change. The subscript 0 is defined in the Eq. 3.

**Modifications**: line 155-157:

We assume the reflectance factor at flight altitude is a good estimation of $BRF_\lambda^e$ just above the surface at infrared wavelengths where atmospheric scattering is negligible.

Eq. 4: the subscript '0' should be defined at this point.

Author's response:

The subscript 0 is defined at line 124.
* * *
**#10 Comment to the Author**

Sect. 3: Subheadings would improve the readability considerably. I recommend to start with some more details about the ARCTAS spring campaign, adding a map with the flight tracks of the measurements used in this study, before giving details about the CAR instrument and the ozone and nitrogen dioxide data.

**Author's response:**

Thanks for pointing this out, yes it will definitely give a better introduction of the data we are using. We added a map of flight track and one paragraph about the campaign.

**Modifications:** line 161-166:

For this study, we used CAR data from the ARCTAS campaign conducted at Elson Lagoon, near Barrow/Utqiaġvik, Alaska, in April 2008 as part of the International Polar Year (Lyapustin et al., 2010; Gatebe and King, 2016). The goal of ARCTAS was to study physical and chemical processes in the Arctic atmosphere (e.g. long-range transport of pollution to the Arctic) and surface parameters (e.g. snow reflectance angular variation). The P-3B aircraft carried CAR instrument and was deployed by NASA from Fairbank. Fig. 1 shows the flight track on 7th of April 2008.

We added this figure as the flight track on 7th of April 2008.

[Figure]

Figure 1: Flight track of P-3B airplane carrying CAR on 07.04.2008 during ARCTAS campaign (Credit: NASA).
* * *
**#11 Comment to the Author**

Figure 1: please add the position of the Sun in the caption of the figure to make it immediately clear where the forward and backward scattering directions are.

**Author's response:**

We added the position of sun to the caption of Fig.2.

**Modifications:** line 793.

… solar zenith angle is 70.23°, 69.11° and 67.78° for flight altitude of 206, 647 and 1700 m respectively.
* * *
**#12 Comment to the Author**

Figure 2: missing whitespace before 1.649 um in the figure caption. The y axis should be named 'Reflectance factor', as this is what you calculate from Eq. 4.

**Author's response:**

Done. We corrected the caption and updated the figure.

Both Figure 1 and 2 should be described in more detail and not only mentioned in the text.

**Author's response:**

We explained detailed features of these two figures at lines 185-194. Perhaps it was not clear that our explanation is referring to these figures, so we made it clear.

**Modifications**: line 182-190:

Here is the explanation:

Examples of calculated reflectance factor values using Eq. (4) from CAR measurements on 7[th] of April, 2008 at Elson Lagoon (71.3° N, 156.4° W) are shown in Fig. 2 and Fig. 3. As we can see in these two figures, in spite of the influence of the atmospheric scattering and absorption, the general features of the snow BRF are clearly observable in polar plots as well as principal and cross plane plots: i) the decrease of snow reflectance with increasing wavelength due to the increasing absorption by snow at longer wavelengths; ii) the increase of the snow BRF as a function of VZA and the strong forward scattering peak in the principal plane at large VZA; iii) the smaller angular variation of the BRF at cross plane compared to the principal plane, though the reflectance values increase with VZA. The snow surface spatial inhomogeneity decreases with increasing altitude due to the change of spatial resolution with altitude (Gatebe and King, 2016; Lyapustin et al., 2010). Accordingly, at poorer spatial resolution, spatial homogeneity are more efficiently averaged as can be seen in Fig. 2 at flight altitude of 1700 m compared to 206 m in which we have higher spatial resolution.
* * *
**#13 Comment to the Author**

P6L173: You are giving an explanation for the decrease in inhomogeneities in the BRDF data. Please also discuss the increase of the BRF with altitude.

**Author's response:**

Sorry, it seems there was a misplacement of words. We should correct this sentence as following:

Modifications: line 186:

...the smaller angular variation of the BRF at cross plane compared to the principal plane, though the reflectance values increase with VZA.
* * *
**#14 Comment to the Author**

P6L182: In the paragraph describing the AOT data, a quick description of the representativeness of the aerosol conditions during the ARCTAS spring campaign with respect to the Barrow climatology would be helpful.

**Author's response:**

Thank you for reminding this point to us, yes its explanation will improve our introduction about the aerosol condition in this area. We added more information about the expected aerosol in Barrow.

**Modifications**: line 197-205:

Aerosol condition and its chemical and optical properties have been measured continuously at Barrow, Alaska, during different seasonal periods (Quinn et al., 2002). Previous studies indicate the largest contribution from sea salt, non-sea-salt sulfate and mineral dust. The average contribution of black carbon is very small compared to other aerosol types (Udisti et al., 2020). During the haze season (January to April), sea salt plays the dominant role in controlling light scattering in wintertime and non-sea salt sulfate in spring (Quinn et al., 2002). The increase on nss-sulfate in January to May is the long-range transport of anthropogenic primary nss sulfate besides the long-range transport of anthropogenic SO2 and its photo-oxidization to nss-sulfate with increase of light levels, and the local production of biogenic nss-sulfate.

In references:

Udisti, R., Traversi, R., Becagli, S., Tomasi, C., Mazzola, M., Lupi, A., and Quinn, P. K.: Arctic Aerosols in: Physics and Chemistry of the Arctic Atmosphere, edited by: Kokhanovsky, A. A., Tomasi, C., Springer Nature Switzerland AG, Cham, Switzerland, 2020.

Quinn, P. K., Miller, T. L., Bates, T. S., Ogren, J. A., Andrews, E., & Shaw, G. E.: A 3-year record of simultaneously measured aerosol chemical and optical properties at Barrow, Alaska., J. Geophys. Res., Atmos., 107(D11), AAC 8-1–AAC 8-15, 2002.
* * *
**#15 Comment to the Author**

P6L183: please provide some more details about the spaceborne measurements of total column ozone.

**Author's response:**

We added more details.

**Modifications**: line 208-210:

This data set (covering from 1995-present) consists of merged total ozone column data retrieved by WFDOAS from Global Ozone Monitoring Experiment (GOME), Scanning Imaging Absorption Spectrometer for Atmospheric Chartography (SCIAMACHY), and GOME-2A.
* * *
**#16 Comment to the Author**

P7L200: I guess the measurement location is sufficiently remote to justify this assumption. However, are there any measurements of black carbon on snow available for this region to further provide evidence for this?

**Author's response:**

Unfortunately, there was no measurement of the snow impurities such as black carbon collocated to the CAR measurements. Nevertheless, to understand the conditions at Barrow better and having a picture of existing aerosol there, we tried to collect information by looking at:

1) Long term continuous measurements of chemical and optical properties of aerosol at Barrow, Alaska (Kokhanovsky and Tomasi, 2020: chapter 4: Udisti et al., 2020): Based on 3 years continuous measurements, during the haze season (January to April), sea salt plays the dominant role in controlling light scattering in wintertime, and non-sea salt sulfate in spring. As can be seen from the Fig. A below, the average contribution of black carbon is very small compared to other aerosol types at Barrow.

2) Aeronet station at Barrow: Although from Aeronet we could not have information about the chemical composition of aerosol, the AOD values before old snow case (7th of April) does not show a significant episodic aerosol event by which snow could be polluted significantly. Our second case study (15th of April) is over fresh fallen snow in which the possibility of being affected by pollutants is even less.

[Figure]

Fig. A: Average composition diagrams of the ground-level particulate matter sampled at Barrow during different seasonal periods. Summer time (June-September): (a), (b), (c): sub-micrometric, super micrometric and overall aerosol particles sampled; In winter time (October-May):(d), (e), (f): sub-micrometric, super-micrometric and overall aerosol particles sampled respectively. Different colours are used to indicate the main particulate matter constituents (sea salt, nss-sulfate and nitrate, mineral dust, black carbon (BC), and water-soluble organic matter (WSOM)

**Modifications**: line 197-205:

Aerosol condition and its chemical and optical properties have been measured continuously at Barrow, Alaska, during different seasonal periods (Quinn et al., 2002). Previous studies indicate the largest contribution from sea salt, non-sea-salt sulfate and mineral dust. The average contribution of black carbon is very small compared to other aerosol types (Udisti et al., 2020). During the haze season (January to April), sea salt plays the dominant role in controlling light scattering in wintertime and non-sea salt sulfate in spring (Quinn et al., 2002). The increase on nss-sulfate in January to May is the long-range transport of anthropogenic primary nss sulfate besides the long-range transport of anthropogenic SO2 and its photo-oxidization to nss-sulfate with increase of light levels, and the local production of biogenic nss-sulfate.

Udisti, R., Traversi, R., Becagli, S., Tomasi, C., Mazzola, M., Lupi, A., and Quinn, P. K.: Arctic Aerosols in: Physics and Chemistry of the Arctic Atmosphere, edited by: Kokhanovsky, A. A., Tomasi, C., Springer Nature Switzerland AG, Cham, Switzerland, 2020.

Quinn, P. K., Miller, T. L., Bates, T. S., Ogren, J. A., Andrews, E., & Shaw, G. E.: A 3-year record of simultaneously measured aerosol chemical and optical properties at Barrow, Alaska., J. Geophys. Res., Atmos., 107(D11), AAC 8-1–AAC 8-15, 2002.
* * *
**#17 Comment to the Author**

P8L229: please provide more details about the 'exponential vertical distribution' used for the vertical profile of the aerosol number density. Are you assuming the aerosol number density is reduced exponentially with height? Is this not influenced by the boundary layer height?

**Author's response:**

The reflectance of a surface-atmosphere system in spectral ranges without strong contribution of gaseous absorption depends mainly on AOT but not on the vertical distribution of aerosol number density. However, to perform radiative transfer calculations one need to assume some number density profile. The exponential profile was selected because it can be used as an approximation in the case of clean aerosol conditions. (see L. Mei, V. Rozanov, M. Vountas, J. P. Burrows, RC.Levy, W. Lotz, Retrieval of aerosol optical properties using MERIS observations: Algorithm and some first results, Remote Sensing of Environment, Volume 197, August 2017, Pages 125-140, for details).

And why were 3 km chosen when the measurements were conducted at flight altitudes below 1700 m?

**Author's response:**

The aerosol above flight altitude affects the downward solar radiation.

Also: for the vertical profiles of pressure and temperature, did you use monthly mean profiles as well or could you make use of radiosonde launches in the vicinity of the study area?

**Author's response:**

We took temperature and pressure profiles from the 2D chemical transport model: Sinnhuber B-M, Sheode N, Sinnhuber M, Chipperfield MP, Feng W , The contribution of anthropogenic bromine emissions to past stratospheric ozone trends: a modelling study. Atmos Chem Phys 2009;9(8):2863–71.

**Modifications**: -
* * *
**#18 Comment to the Author**

Figure 3: (1) The ice crystal shapes presented in Figure 3 do not match the 9 morphologies introduced on P7L206: it seems you are presenting solid bullet rosettes in the figure, which are not mentioned in the text. On the other hand, you are not presenting the results for the fractal particles. Please clarify that as it is a bit confusing to me.

**Author's response:**

Sorry for the confusion. We added solid bollet rosette to text as well. Unfortunately, the database for solid bullet rosette was not fully ready in SCIATRAN. And we could not use it for effective radius retrieval. However, we could calculate and show the effect of its shape and size in Fig. 3.

Fractal is not shown in Fig.3, because in this figure, we focus on the new database of SCIATRAN and fractal was an old snow morphology. In addition, the range and existing size interval of fractal is completely different from the new database and ice crystals and we think having fractal in this figure will raise more confusion for the reader.

**Modifications:-**
* * *
**#19 Comment to the Author**

Figure 3: I assume this is still the calculated Reflectance factor, please name the y axis accordingly.

**Author's response:**

We corrected the label of Y axis accordingly. Now it is figure 4.

**Modifications:**

Please see figure 4.
* * *
**#20 Comment to the Author**

P9L264: please specify 'same size', as in the sentence before you are talking about a size range between 60 to 10000 um.

**Author's response:**

We clarified this sentence.

**Modifications**: line 291-294

If we change only the shape of snow grain from "aggregate of 8 columns" to the "droxtal", but we keep the size (largest dimension) as it is (e.g. 300 μm) this change provides a noticeable decrease of ~ 30% in reflectance at forward scattering direction for a viewing zenith angle of 60° and leads to a much weaker forward peak.
* * *
**#21 Comment to the Author**

P9L266: please clarify this sentence, because when I look at Figure 3, also for the plate shape the reflectance factor increases in the backward direction compared to the nadir direction.

**Author's response:**

We clarified it. We meant in comparison to other shapes such as the aggregate of 8 column shape or droxtal.

**Modifications**: line 294-295

Noteworthy is, that the plate shape cannot reproduce the enhancement in backward direction (typical for a BRF over snow) as strong as the "aggregate of 8 columns" or the "droxtal" shape cause.
* * *
**#22 Comment to the Author**

P9L267: larger reflectance in all directions compared to what? The reflectance factor for the hollow bullet rosette seems to be at least equally high for some snow grain sizes compared to the aggregates of 5 or 10 plates.

**Author's response:**

We corrected the sentence.

**Modifications**: line 296-297

Using the "aggregate of 5 and 10 plates" leads to larger reflectance in all directions compared to the single "plate" shape.
* * *
**#23 Comment to the Author**

Figure 4: (1) the green and blue lines and too similar and are hard to distinguish within the plot. (2) this is a reflectance factor again? Please name the y axis accordingly.

**Author's response:**

Sorry for the color, we corrected it and changed the label of Y axis.

**Modifications:**

Please see the figure:
* * *
[Figure]
* * *
**#24 Comment to the Author**

Figures 5 and 6: (1) this is a reflectance factor again? Please name the y axis accordingly.

**Author's response:**

We corrected the Y axis label.

**Modifications:**

Please see Fig. 6 and 7.
* * *
**#25 Comment to the Author**

 Figures 5 and 6: (2) The uncertainty of the CAR measurements needs to be included in the figure in the form of error bars. This also needs to be considered when calculating the RMSE. I assume a difference in RMSE of less than 0.4 % as visible between the chosen aggregates of 8 columns (98.8 um) and the columns (74.7 um) is not

significant when considering possible measurement and retrieval uncertainties. This needs to be discussed in the manuscript.

**Author's response:**

We agree that uncertainty f measurements, simulation and retrieval are very important and should be plotted and discussed (as we explained in the beginning of comments). We added uncertainty envelopes to measurements in Fig. 5, 6 (now they are 6 and 7), which is 5% based on a comprehensive study done in NASA in 2007. We also provided this information in the text (as explained before).

Modifications:

Please see Fig. 6 and 7 and the text.
* * *
**#26 Comment to the Author**

The surface roughness clearly affects the CAR measurements at large viewing zenith angles. As I understand, the macroscopic surface roughness (in contrast to the ice crystal roughness) is not included in the SCIATRAN simulations?

In this case, I suspect you are trying to fit the simulations to the measurements using different single scattering properties for the different ice crystal shapes, while more probably the macroscopic surface roughness is the underlying reason for the deviations between CAR measurements and SCIATRAN simulations. Macroscopic surface roughness enhances the backscatter by changing the effective angle of incidence, and reduces the forward scatter by casting shadows. Of course, this depends on the size of the roughness structures and their orientation, and I guess both parameters are unknown for the measurement conditions? Maybe some observations from within the aircraft with the naked eye or camera pictures could give an indication? At least the reduction in forward scattering of the CAR measurements compared to the simulations is visible for many different ice crystal shapes in Figure 5. Figure 6, however, shows an increase in the forward scattering as measured with CAR. In trying to choose the lowest RMSE for model simulations that neglect macroscopic surface roughness, it seems to me you can partly mimic the effect of surface roughness in choosing different ice crystal shapes (and single scattering properties). Thus, you are getting the 'right simulation', but for the wrong reasons in my point of view. Is there any way to test your simulations for different macroscopic surface roughness heights and orientation? Either way, this uncertainty needs to be discussed in detail within the manuscript.

**Author's response:**

Unfortunately, as you wrote macroscopic surface roughness is not included in the SCIATRAN. However, in Fig. 5 and 6, we are looking to 1.6 μm. We think at this wavelength, the effect of shadowing because of surface

inhomogeneity is minimal. We do not think our effective radius retrieval and forward modeling for reflectance calculation at this spectral range is that much affected by surface roughness.

When we move to shorter wavelengths, such as 0.6 and 0.8 μm (Fig 8 and 9), we see the difference between measurement and simulation gets larger. We mentioned in the text that this larger difference can be due to surface roughness.

About the possible surface roughness in the measurement area, we know that we have sastrugi (with ~5cm height) in fresh snow case (we don't know about old snow), but no precise information about their orientation. But because of not modeling it in SCIATRAN, we could not estimate its effect unfortunately.

**Modifications: -**
* * *
**#27 Comment to the Author**

P11L333-345: The justification of the ice crystal shape retrievals with the temperature dependence seems dubious to me. One needs to be careful in differentiating the important temperatures here. It is true that temperature (and supersaturation!) strongly affect the shape of pristine ice crystals when the precipitating snow is formed within the cloud. If anything, the ice crystal shape should be connected to the temperature profile at the time of the last snowfall (excluding snow aging processes). However, the temperatures you are stating are temperatures measured in-flight, probably days after the precipitation event. This temperature is completely unrelated to the snow on the ground, especially as you report yourself that the snow surface consists of old snow during most days. After the snow has fallen to the ground, the vertical temperature gradient at the surface and within the snowpack is way more important for the ice crystal shape (influencing snow metamorphism processes). If you don't have in situ observations looking at the ice crystal shape on the ground, you cannot validate your ice crystal shape retrieval in that way.

**Author's response:**

We agree that temperature and super-saturation effect needs more investigation, which was not in the scope of our manuscript. However, we mentioned a few sentences about the possible relation between snow morphology and the temperature to highlight that there is a room for investigation. Though, we did not use temperature argument to validate our findings about snow morphology, we understand your point, we should not bring it in the abstract and so we delete it from abstract because it's not a confirmed finding of our work.

But we kept a few sentences in the discussion part. And we added your point that one needs to be careful about temperature and emphasized that more investigation is needed.

We think the temperature is representative enough for fresh snow case but we agree that for old snow case, the temperature at the time of snowfall is also important. We added your point to the paper.

If you think it's fine to keep it after the mentioned modification in the manuscript we would do so, if not, we will delete it.

**Modifications**: line 362-372:

Though the real nature of ice crystal shape at the time of measurement is not known to us, the impact of temperature and supersaturation on morphology of snow grain particles has been debated in previous studies. (Slater and Michaelides, 2019; Shultz, 2018; Libbrecht, 2007; Bailey and Hallett, 2004; Yang et al., 2003). Based on the relationship between temperature and snow grain morphology, the column-based shapes are the dominant ice crystal morphology in environments with temperatures higher than -10°C whereas plates are dominant if the temperature is less than -10°C. Though, more investigation is needed especially to account for the temperature profile at the exact time of snowfall, our findings with respect to the most representative shape for each case study agree with this argument. The temperature range during CAR measurements at 6-7th of April 2008 is from -20 to -5°C. Based on our results the "aggregate of 8 columns" is the most representative shape for measurements conducted on this day. On 15th of April 2008 when the temperature range changes to -23 to -17°C, mainly plate-based ice crystal shapes are expected for such low temperatures and our results confirm this argument.
* * *
**#28 Comment to the Author**

Figure 7: This is way more illustrative and provides more information than Table 2, which becomes redundant in my point of view and can be removed from the manuscript.

**Author's response:**

Referee Nr. 1 asked us to add asymmetry parameter to table 2 (now is table 3). So we still tend to keep it if it is fine. Please let us know if you think we should remove it.

**Modifications**:-
* * *
**#29 Comment to the Author**

P12L359: it seems you are normalizing the RMSE somehow. Please provide the formula how you calculated the RMSE, as your description in the text seems to be imprecise.

**Author's response:**

To calculate RMSE:

$$RMSE=\sqrt{\frac{1}{n}\sum_{i=1}^{n}\left(\frac{x_{obs}-x_{simulation}}{x_{obs}}\right)^2}\times 100$$

**Modifications:-**
* * *
**#30 Comment to the Author**

P12L364: please round the effective radii to integer values. Providing two decimals is implying a degree of accuracy which is not achieved.

**Author's response:**

done.

**Modifications**: line 398-400

…with a maximum dimension of 650 µm (effective radius 99 µm) for the case of old snow, and "aggregate of 5 plates" ice crystals with a maximum dimension of 725 µm (effective radius 83 µm) for the case of fresh snow.
* * *
**#31 Comment to the Author**

Figure 8: caption: 'reflectance factor'

**Author's response:**

Done.

**Modifications**: caption of Fig . 9:

Left column shows reflectance factor at three wavelengths:…
* * *
**#32 Comment to the Author**

P12L367: I would recommend introducing Figure 10 only after Figures 8 and 9.

**Author's response:**

We moved the introduction of this figure to after Figure 8 and 9.

**Modifications**: line 402-410:

In Fig. 9, the difference between the simulated and measured reflectance factor at 0.677 and 1.649 µm is small on average, being less than 0.025 in regions of small VZA and not exceeding ±0.05 for larger VZA < 50°. These values are larger at 0.873 µm; the maximum difference reaches ~ ±0.05 for small VZA. The difference between SCIATRAN simulation values and those of the measurements is pronounced in the forward scattering region where |Δφ| < 40°. Fig. 10 is the same plot as Fig. 9 but for fresh snow. The differences between SCIATRAN simulations and CAR

measurements of the reflectance factor are less pronounced in the glint region, as compared to those for the old snow.

To assess the accuracy of simulations over all azimuth angles, the correlation plot and the Pearson correlation coefficient between measured and modelled reflectance factor are shown in Fig. 11. As it is shown in …
* * *
**#33 Comment to the Author**

Figure 10: (1) caption: 'reflectance factor', x and y axis: 'reflectance factor', please state again in the caption which columns belong to the old and new snow cases.

**Author's response:**

Done.

**Modifications:** Figure 11 and caption:

Caption: The scatter plot with corresponding pearson correlation coefficient of reflectance factor measured by CAR and simulated by SCIATRAN; left column shows he results for old snow, right column: fresh snow. Here the color bar represents number density of pixels.
* * *
**#34 Comment to the Author**

I am interested in seeing a comparison of the correlation coefficients between new snow case and the lowest flight level of the old snow case as they have roughly comparable flight altitudes. This might make it easier to discuss a possible influence of surface inhomogeneities. At this point it would also help to provide more details about the differences in flight tracks between the two measurement days. Was the same area probed on both days? Otherwise of course, even the same flight altitude might not be comparable. This is connected to my earlier comment to provide more details about the actual flights performed during the campaign.

**Author's response:**

Unfortunately, the case of fresh and old snow in our study are not exactly over the same area (We have the coordinates of flight path and measurement). Actually, we were very much interested to see this comparison to understand more about the effect of surface inhomogeneity. Nevertheless, we were not able to do so because they are not comparable.

**Modifications**: –
* * *
**#35 Comment to the Author**

P12L377: I do not agree with the conclusion drawn here. The high correlation coefficient and small discrepancies do not justify the selection of this wavelength channel for the selection of the best ice crystal shape. The correlation coefficient and small bias is made 'by construction', as you selected the ice crystal shape based on the lowest bias between simulation and CAR measurements in the first place. The high correlation coefficients for this wavelength channel are therefore not surprising.

**Author's response:**

We agree, and deleted this sentence.

**Modifications:** line 416.
* * *
**#36 Comment to the Author**

The last sentence of this section (P12L379) seems a bit out of place and should be rephrased.

**Author's response:**

Sorry this sentence should move to previous paragraph in which we compare polar plots. It was misplaced during several modifications of our paper.

**Modifications:** moved from line 418 to line 407.

Please see text.
* * *
**#37 Comment to the Author**

P13L388: This is an important point and should be included in this study already by looking at the correlation coefficient between measured and simulated reflectance factors and their dependence on the flight altitude for the case of old snow. I am interested to see whether there is a clear dependence of the correlation coefficient on the flight altitude.

**Author's response:**

We calculated the correlation coefficient for Fig. 5 which is a comparison of reflectance at different altitudes at 0.67μm, principal plane. The correlations gets higher when we move from ~200 m to 600 m. But there wasn't an increase when we move from 600m altitude to 1700m.

**Modifications:-**
* * *
**#38 Comment to the Author**

P13L405: With regard to my earlier comment, the justification of the ice crystal shape retrieval with the temperature dependence cannot be mentioned here (and also not in the abstract).

**Author's response:**

We explained in previous comment that we did not use temperature argument to validate our findings. The aim of mentioning this relation between snow morphology and temperature was more to highlight that there is room to investigate the possible relation. But we understand your point, we deleted this argument from abstract (as it needs more investigation and it's better not to represent it in the abstract). But we kept a few sentences in discussion part. And we added your point that one needs to be careful about temperature and emphasized that more investigation is needed.

If you think it's fine to keep it after the mentioned modification in the manuscript we would do so, if not, we will delete it.

**Modifications**: line 362-372:

Though the real nature of ice crystal shape at the time of measurement is not known to us, the impact of temperature and supersaturation on morphology of snow grain particles has been debated in previous studies. (Slater and Michaelides, 2019; Shultz, 2018; Libbrecht, 2007; Bailey and Hallett, 2004; Yang et al., 2003). Based on the relationship between temperature and snow grain morphology, the column-based shapes are the dominant ice crystal morphology in environments with temperatures higher than -10°C whereas plates are dominant if the temperature is less than -10°C. Though, more investigation is needed especially to account for the temperature profile at the exact time of snowfall, our findings with respect to the most representative shape for each case study agree with this argument. The temperature range during CAR measurements at 6-7[th] of April 2008 is from -20 to -5°C. Based on our results the "aggregate of 8 columns" is the most representative shape for measurements conducted on this day. On 15[th] of April 2008 when the temperature range changes to -23 to -17°C, mainly plate-based ice crystal shapes are expected for such low temperatures and our results confirm this argument.
* * *
**#39 Comment to the Author**

P13L416: I wonder why the use of a vertically inhomogeneous snow layer in the model is only mentioned here and not in the discussion of the results already. It should not be mentioned for the first time in the Conclusions in my point of view.

**Author's response:**

Assuming vertically inhomogeneous snow layer and investigating its effect was not in the scope of our work and we mentioned it only in the conclusion as a room to improve and consider in future works. That is why we did not mention this in the discussion of results. We hope it's fine to keep it as it is because we do not have enough information to present about it in the discussion section.

**Modifications**: -

**Technical corrections**

**#1 Comment to the Author**

P1L30: Assuming that the snow layer consists [...]

**Author's response:** Done.

**Modifications**: line 25: Assuming that the snow layer consists [...]
* * *
**#2 Comment to the Author**

P3L78: delete ';' after '2011'

**Author's response:** Done.

**Modifications:** line 81...Kokhanovsky, 2011).
* * *
**#3 Comment to the Author**

P3L88: 'phenomenological', 'airborne'

**Author's response:** Done.

**Modifications:** line 92-93...well validated phenomenological RTM, and the airborne observations...
* * *
**#4 Comment to the Author**

P5L126: of the surface

**Author's response:** Done.

**Modifications**: line 127: To isolate the reflectance properties of the surface...

**#5 Comment to the Author**

P5L127: on the measured radiance

**Author's response**: Done.

**Modifications:** line 128:…correction methods on the measured radiance at TOA…
* * *
**#6 Comment to the Author**

P5L128: scattering or absorption applying RTMs

**Author's response**: Done.

**Modifications**: line 129:…scattering or absorption applying RTMs…
* * *
**#7 Comment to the Author**

P5L128: This removes the four atmospheric […] from the measured radiance: i) […]

**Author's response**: Done.

**Modifications**: line 129: This removes the four atmospheric contributions from the measured radiance at TOA or flight altitude:…
* * *
**#8 Comment to the Author**

P5L153: delete 'etc' or be more specific

**Author's response:** Done.

**Modifications:** line 160:…up to present to measure the single scattering albedo of clouds, the bidirectional reflectance of various surface types and acquiring imagery of clouds and the Earth's surface.
* * *
**#9 Comment to the Author**

P6L160: by a mirror – missing blank

**Author's response**: Done.

**Modifications:** line 171: CAR collects data by a mirror rotating 360° in a plane perpendicular…
* * *
**#10 Comment to the Author**

P6L163: do you mean viewing zenith and azimuth angles?

**Author's response:** Yes, thanks, done.

**Modifications**: line 174:...both viewing zenith and azimuth angles...
* * *
**#11 Comment to the Author**

P6L163: Please rephrase: The high [...] resolution [...] allows the estimation of the anisotropy of the reflectance in the snow-atmosphere system with high accuracy.

Author's response: Thanks, done.

**Modifications:** line 174: The high angular/spatial resolution of 1° in both viewing zenith and azimuth angles allows the estimation of the anisotropy of the reflectance in the snow-atmosphere system with high accuracy.
* * *
**#12 Comment to the Author**

P6L167: RTM simulations

Author's response: Done.

**Modifications:** line 180:...atmospheric effects in RTM simulations.
* * *
**#13 Comment to the Author**

P6L175: do you mean spatial inhomogeneity?

**Author's response**: Yes. Done.

**Modifications:** line 187: The snow surface spatial inhomogeneity decreases...
* * *
**#14 Comment to the Author**

P7L208: eight ice crystal shapes/habits

**Author's response**: Done.

**Modifications:** line 229: Recently, a new data library of basic single scattering properties of eight ice crystal shapes/habits developed by...
* * *
**#15 Comment to the Author**

P7L208: (referred to as fractal in this paper)

**Author's response**: Done.

**Modifications:** line 235: Koch fractal (referred to as fractal in this paper) particles are used as well

**#16 Comment to the Author**

P8L227: ground-based measurements from AERONET

**Author's response**: Done.

**Modifications:** line 253: Using AOT from ground-based measurements of AERONET…
* * *
**#17 Comment to the Author**

P8L228: selecting one of the aerosol types

**Author's response**: Done.

**Modifications:** line 254:…as mentioned and selecting one of the aerosol types…
* * *
**#18 Comment to the Author**

P9L271: a priori knowledge

**Author's response:** Done.

**Modifications:** line 300: … highlights the importance of having accurate a priori knowledge or estimation of size of the ice crystals and their shapes to simulate accurately measurements.
* * *
**#19 Comment to the Author**

P9L272: to accurately reproduce measurements

**Author's response**: Done.

**Modifications:** line 300:… a priori knowledge or estimation of size of the ice crystals and their shapes to accurately reproduce measurements.
* * *
**#20 Comment to the Author**

P9L279: evaluate the impact of the atmosphere

**Author's response:** Done.

**Modifications:** line 309: Therefore, in this section, absorption bands e.g. 0.677 μm are selected to evaluate the impact of the atmosphere.
* * *
**#21 Comment to the Author**

P9L282: assuming the following properties

**Author's response**: Done.

**Modifications:** line 312: The calculations are performed assuming the following properties of the atmosphere and snow layer:...
* * *
**#22 Comment to the Author**

P12L362: described in the previous sections

**Author's response**: Done.

**Modifications**: line 396: The simulations, which used the results and findings described in the previous section were performed...
* * *
**#23 Comment to the Author**

P12L380: measurement of the reflectance factor

**Author's response:** Done.

**Modifications:** line 407: The differences between SCIATRAN simulations and CAR measurements of the reflectance factor are less pronounced...
* * *
**#24 Comment to the Author**

 P13L390: reflectance factor

**Author's response**: Done

**Modifications:** line 429: The SCIATRAN RTM (a phenomenological RTM) was used to simulate the reflectance factor in the snow-atmosphere...
* * *
**#25 Comment to the Author**

P13L397: reflectance factor

**Author's response:** Done.

**Modifications**: line 436, In our case study at Barrow/Utqiaġvik, the simulated reflectance factor assuming…
* * *
**#26 Comment to the Author**

P13L408: reflectance factor

**Author's response**: Done.

**Modifications:** line 447: In our study, the simulated patterns of the reflectance factor with respect to spectral and directional signatures produce well the measurements...
* * *
**#27 Comment to the Author**

P13L411: reflectance factor

**Author's response:** Done.

**Modifications:** line 450:... the overall absolute difference between the modeled reflectance factor from SCIATRAN and CAR...
* * *
**#28 Comment to the Author**

P14L429: comma misplaced

**Author's response**: Done.

**Modifications:** line 468... respectively.
* * *
**#29 Comment to the Author**

P15L466: do you really mean TOA reflectance? Or reflectance at flight altitude?

**Author's response**: Thanks for pointing it out, we should say reflectance at flight altitude. Done.

[revised manuscript text omitted]

---

## Author Comment (AC4)

**Reply to Referee #1**

Dear Dr. Kokhanovsky,

We would like to thank you for the review and your constructive comments which helped to improve our manuscript.

Our point-by-point responses to the specific comments (in red) are given in blue and the modification made in the manuscript is presented in green. This document also includes a marked-up version of manuscript.

Best Regards,

Soheila Jafariserajehlou
* * *
**#1 Comment to the Author:**

The selection of the appropriate snow grain shape and size must be performed using both angular and spectral measurements. The authors discuss mainly the angular patterns. It is interesting to see how differ spectral reflectances for different best models shown in Table 2 and how they agree with spectral reflectance measurements at 14 Cloud Absorption Radiometer (CAR) spectral channels.

**Author's response:**

This is definitely true; it would be very interesting to see the change of reflectance in the wide spectral range of CAR. Unfortunately, the investigation of all wavelengths was not considered in the scope of our study mainly because of two problems:

1. The variation of ice absorption with the increase of wavelength in the near IR range: This means that the photon penetration depth in snow layer will depend on the wavelength. Considering our assumption in SCIATRAN: a vertically homogeneous snow layer, we will obtain for different wavelengths different effective radius of ice crystals. With these results, we could only confirm that a snow layer is vertically in-homogeneous. However, we know this without additional calculations. Therefore, we would consider the reflectance at wavelengths in more near IR range only if we could assume vertically in-homogeneous snow layer and formulate the inverse problem with respect to the vertical profile of ice crystal effective radius.

However, the consideration of this problem and solution of its inverse problem is out of scope of this manuscript but we will definitely consider it in the future studies.

2. Not all of the 14 spectral channels of CAR were active during all of our measurements.

**Modifications**: -
* * *
**#2 Comment to the Author:**

Please, list the CAR channels in the paper.

**Author's response:**

Thanks for reminding this point to us. We added a table including all channels of CAR and their bandwidth.

**Modification:** Table 2. Summary of CAR wavelengths and bandwidth.

| Channel number | Central wavelengths in µm | Bandwidth in nm |
|---|---|---|
| 1 | 0.480 | 21 |
| 2 | 0.687 | 26 |
| 3 | 0.340 | 9 |
| 4 | 0.381 | 6 |
| 5 | 0.870 | 10 |
| 6 | 1.028 | 4 |
| 7 | 0.609 | 9 |
| 8 | 1.275 | 24 |
| 9 | 1.554 | 33 |
| 10 | 1.644 | 46 |
| 11 | 1.713 | 46 |
| 12 | 2.116 | 43 |
| 13 | 2.203 | 43 |
| 14 | 2.324 | 48 |

**#3 Comment to the Author:**

Asymmetry parameters in the visible must be given for all cases shown in Table 2.

Author's response: Done. We added asymmetry parameter to table 2 (now is 3).

**Modification:** table 3.

| Ice crystal habit | Asymmetry parameter Old snow | Fresh snow | | Retrieved effective radius (µm) Old snow | Fresh snow | | Old snow Bias (%) | RMSE (%) | | Fresh snow Bias (%) | RMSE (%) | |
|---|---|---|---|---|---|---|---|---|
| Fractal | 0.825 | 0.827 | 69.37 | 76.06 | 3.50 | 9.75 | 13.16 | 14.69 |
| Droxtal | 0.856 | 0.863 | 94.48 | 106.95 | 0.87 | 25.54 | 10.10 | 34.14 |
| Column | 0.873 | 0.877 | 74.71 | 80.49 | 2.17 | 7.32 | 12.36 | 15.72 |
| Hollow column | 0.884 | 0.888 | 67.32 | 72.85 | 2.80 | 11.15 | 13.66 | 15.14 |
| Aggregate of 8 columns | 0.844 | 0.849 | 98.83 | 107.62 | 2.79 | 6.97 | 11.85 | 18.27 |
| Plate | 0.923 | 0.942 | 38.93 | 61.44 | -0.44 | 21.47 | 11.68 | 16.99 |
| Aggregate of 5 plates | 0.874 | 0.877 | 78.02 | 83.41 | 1.82 | 10.34 | 11.23 | 12.85 |
| Aggregate of 10 plates | 0.893 | 0.893 | 65.36 | 69.28 | 2.34 | 13.91 | 11.52 | 13.16 |
| Hollow-bullet rosette | 0.887 | 0.889 | 67.01 | 73.28 | 2.16 | 9.99 | 12.71 | 15.16 |

**#4 Comment to the Author:**

The authors assume clean snow. I think, the authors must show some evidence in the paper that the measured spectra have not been affected by possible snow pollution.

**Author's response:**

Unfortunately, there was no measurement of the snow impurities such as black carbon collocated to the CAR measurements. Nevertheless, to understand the conditions at Barrow better and having a picture of existing aerosol there, we tried to collect information by looking at:

1) Long term continuous measurements of chemical and optical properties of aerosol at Barrow, Alaska (Kokhanovsky and Tomasi, 2020: chapter 4: Udisti et al., 2020): Based on 3 years continuous measurements, during the haze season (January to April), sea salt plays the dominant role in controlling light scattering in wintertime, and non-sea salt sulfate in spring. As can be seen from the Fig. A below, the average contribution of black carbon is very small compared to other aerosol types at Barrow.

2) Aeronet station at Barrow: Although from Aeronet we could not have information about the chemical composition of aerosol, the AOD values before old snow case (7[th] of April) does not show a significant episodic aerosol event by which snow could be polluted significantly. Our second case study (15[th] of April) is over fresh fallen snow in which the possibility of being affected by pollutants is even less.

[Figure]

Fig. A: Average composition diagrams of the ground-level particulate matter sampled at Barrow during different seasonal periods. Summer time (June-September): (a), (b), (c): sub-micrometric, super micrometric and overall aerosol particles sampled;

In winter time (October-May):(d), (e), (f): sub-micrometric, super-micrometric and overall aerosol particles sampled respectively. Different colours are used to indicate the main particulate matter constituents (sea salt, nss-sulfate and nitrate, mineral dust, black carbon (BC), and water-soluble organic matter (WSOM).

**Modifications:** line 201-210.

Aerosol condition and its chemical and optical properties have been measured continuously at Barrow, Alaska, during different seasonal periods (Quinn et al., 2002). Previous studies indicate the largest contribution from sea salt, non-sea-salt sulfate and mineral dust. The average contribution of black carbon is very small compared to other aerosol types (Udisti et al., 2020). The increase on nss-sulfate in January to May is the long-range transport of anthropogenic primary nss sulfate besides the long-range transport of anthropogenic SO2 and its photo-oxidization to nss-sulfate with increase of light levels, and the local production of biogenic nss-sulfate.

Udisti, R., Traversi, R., Becagli, S., Tomasi, C., Mazzola, M., Lupi, A., and Quinn, P. K.: Arctic Aerosols in: Physics and Chemistry of the Arctic Atmosphere, edited by: Kokhanovsky, A. A., Tomasi, C., Springer Nature Switzerland AG, Cham, Switzerland, 2020.

Quinn, P. K., Miller, T. L., Bates, T. S., Ogren, J. A., Andrews, E., & Shaw, G. E.: A 3-year record of simultaneously measured aerosol chemical and optical properties at Barrow, Alaska., J. Geophys. Res., Atmos., 107(D11), AAC 8-1–AAC 8-15, 2002
* * *
**#5 Comment to the Author:**

Line 4, leads–>lead;

**Author's response**: Done.

**Modification:** (moved to line 51): However, the current differences between simulated and measured reflectance in a coupled snow-atmosphere system, lead to ….
* * *
**#6 Comment to the Author:**

line 32, to be –>is;

**Author's response:** Done.

**Modifications:** line 26…we find that for a surface covered by old snow, the Pearson correlation coefficient, R, between measurements and simulations is 0.98 (R2 ~ 0.96).
* * *
**#7 Comment to the Author:**

line 46 (AA);

**Author's response:** Done.

**Modifications:** line 46. This phenomenon is known as the Arctic Amplification (AA) (Serreze and Barry, 2011).
* * *
**#8 Comment to the Author:**

line 204, is reference available?

**Author's response:**

Yes, when we wrote this manuscript the reference was under preparation but now it is published:

https://www.sciencedirect.com/science/article/abs/pii/S0022407320302442

**Modifications:** line 236 and reference updated at line 620:

Pohl, C., Rozanov, V. V., Mei, L., Burrows, J. P., Heygster, G., Spreen, G.,: Implementation of an extensive ice crystal single-scattering property database in the radiative transfer model SCIATRAN, J. Quant. Spectrosc. Ra., 253, https://doi.org/10.1016/j.jqsrt.2020.107118, 2020.
* * *
**#9 Comment to the Author:**

Line 210, did you assume rough Koch crystals?

**Author's response**:  Yes. We assumed rough Koch crystals.

**Modifications:** -
* * *
**#10 Comment to the Author:**

line 240, the wavelength of 1.24 microns is more suitable for the grain size retrieval ( larger sensitivity to the grain size);

**Author's response:**

In the retrieval algorithm, we assume a vertically homogeneous snow layer. To decrease the photon penetration depth and estimate the effective radius of ice crystals near the top of snow layer the wavelength 1.65 micron with stronger absorption was selected.  The measurement errors in this spectral channel cannot significantly decrease the accuracy of inverse problem solution. Figure 6 of the manuscript clearly demonstrates dependence  of  RMSE on the selected shape and even on roughness of ice crystal. This confirms that the sensitivity of reflectance with respect to the effective radius is high enough.

**Modifications:** -
* * *
**#11 Comment to the Author:**

line 250, matrix->function

**Author's response**: Done.

**Modifications:** line 280:

The BRF properties of snow at 1.649 μm are closer to that for single scattering behavior and it is linked to the phase function, which strongly depends on the shape of ice crystals.
* * *
**#12 Comment to the Author:**

line 271, a priori

**Author's response**: Done.

**Modifications:** line 303:

The large range of changes of the reflectance when using different ice crystal sizes in both the principal and cross planes highlights the importance of having accurate a priori knowledge or estimation of size of the ice crystals and their shapes to simulate accurately measurements.
* * *
**#13 Comment to the Author:**

line 276, remove 'on the snow layer'

**Author's response:** Done.

[revised manuscript text omitted]